# Synaptic mitochondria regulate hair-cell synapse size and function

Hiu-tung C Wong[1,2], Qiuxiang Zhang[1], Alisha J Beirl[1], Ronald S Petralia[3], Ya-Xian Wang[3], Katie Kindt[1]*

[1]Section on Sensory Cell Development and Function, National Institute on Deafness and Other Communication Disorders, National Institutes of Health, Bethesda, United States; [2]National Institutes of Health-Johns Hopkins University Graduate Partnership Program, National Institute on Deafness and Other Communication Disorders, National Institutes of Health, Bethesda, United States; [3]Advanced Imaging Core, National Institute on Deafness and Other Communication Disorders, National Institutes of Health, Bethesda, United States

**Abstract** Sensory hair cells in the ear utilize specialized ribbon synapses. These synapses are defined by electron-dense presynaptic structures called ribbons, composed primarily of the structural protein Ribeye. Previous work has shown that voltage-gated influx of $Ca^{2+}$ through $Ca_V1.3$ channels is critical for hair-cell synapse function and can impede ribbon formation. We show that in mature zebrafish hair cells, evoked presynaptic-$Ca^{2+}$ influx through $Ca_V1.3$ channels initiates mitochondrial-$Ca^{2+}$ (mito-$Ca^{2+}$) uptake adjacent to ribbons. Block of mito-$Ca^{2+}$ uptake in mature cells depresses presynaptic-$Ca^{2+}$ influx and impacts synapse integrity. In developing zebrafish hair cells, mito-$Ca^{2+}$ uptake coincides with spontaneous rises in presynaptic-$Ca^{2+}$ influx. Spontaneous mito-$Ca^{2+}$ loading lowers cellular $NAD^+$/NADH redox and downregulates ribbon size. Direct application of $NAD^+$ or NADH increases or decreases ribbon size respectively, possibly acting through the NAD(H)-binding domain on Ribeye. Our results present a mechanism where presynaptic- and mito-$Ca^{2+}$ couple to confer proper presynaptic function and formation.

*For correspondence:
katie.kindt@nih.gov

Competing interests: The authors declare that no competing interests exist.

## Introduction

Neurotransmission is an energy demanding process that relies heavily on mitochondria. In neurons, mitochondrial dysfunction has been implicated in synaptopathies that impact neurodevelopment, learning and memory, and can contribute to neurodegeneration (*Flippo and Strack, 2017*; *Lepeta et al., 2016*; *Todorova and Blokland, 2016*). In hair cells, sensory neurotransmission relies on specialized ribbon synapses to facilitate rapid and sustained vesicle release that is particularly energy demanding (reviewed in: *Johnson et al., 2019*; *Lagnado and Schmitz, 2015*; *Matthews and Fuchs, 2010*; *Safieddine et al., 2012*). Although mitochondrial dysfunction has been implicated in hearing loss (*Böttger and Schacht, 2013*; *Fischel-Ghodsian et al., 2004*; *Kokotas et al., 2007*), the precise role mitochondria play at hair-cell synapses remains unclear.

Ribbon synapses are characterized by a unique presynaptic structure called a 'ribbon' that tethers and stabilizes synaptic vesicles at the active zone (reviewed in: *Matthews and Fuchs, 2010*). In hair cells, neurotransmission at ribbon synapses requires the presynaptic-$Ca^{2+}$ channel $Ca_V1.3$ (*Brandt et al., 2003*; *Kollmar et al., 1997*; *Sidi et al., 2004*). Hair-cell depolarization opens $Ca_V1.3$ channels, resulting in a spatially restricted increase of $Ca^{2+}$ at presynaptic ribbons that triggers vesicle fusion. Tight spatial regulation of presynaptic $Ca^{2+}$ is important for ribbon-synapse function and requires efficient $Ca^{2+}$ clearance through a combination of $Ca^{2+}$ pumps, $Ca^{2+}$ buffers and intracellular $Ca^{2+}$ stores (*Carafoli, 2011*; *Mulkey and Malenka, 1992*; *Tucker and Fettiplace, 1995*; *Yamoah et al., 1998*; *Zenisek and Matthews, 2000*). While ER-$Ca^{2+}$ stores have been implicated in

**eLife digest** Hearing depends upon specialized cells deep within the ear called hair cells. These cells take their name from the bundles of hair-like fibers found on their surface, which move when sound waves enter the ear. This movement activates the hair cells, which send signals to nearby neurons at contact points called synapses. Hair cells must send messages to their synaptic partners rapidly and continuously in order for humans to follow complex streams of sound, such as speech. This requires large amounts of energy, which are produced by compartments inside the hair cells called mitochondria.

Wong et al. show that mitochondria, which are often described as the 'power plants' of cells, are critical for hair cell synapses to form and work correctly. But rather than studying hair cells in the human ear, Wong et al. took advantage of the fact that another species – the zebrafish – has hair cells on its body surface. These cells detect movements in water rather than sound waves, but they work in much the same way as hair cells in the ear, and are easier to access and study.

Wong et al. report that in zebrafish larvae, developing hair cells send spontaneous signals to their contact neurons even before they start receiving any sensory input. But if mitochondria in the hair cells fail to detect these signals, the synapses fail to form correctly. In older zebrafish, mature hair cells send signals to their synaptic partners whenever they detect sensory input. But if mitochondria fail to detect these signals, the synapses stop working and ultimately break down.

These findings help explain why damage to mitochondria in the inner ear can lead to hearing loss. Moreover, because mitochondria are present in almost all cells, their disruption causes a wide range of diseases. Many of these involve the brain, which requires large amounts of energy and so is particularly vulnerable to mitochondrial damage. These results may provide insights into such disorders.

hair-cell neurotransmission, whether mitochondrial-$Ca^{2+}$ (mito-$Ca^{2+}$) stores play a role in this process remains unclear (*Castellano-Muñoz and Ricci, 2014*; *Kennedy, 2002*; *Lioudyno et al., 2004*; *Tucker and Fettiplace, 1995*).

In addition to a role in hair-cell neurotransmission, presynaptic $Ca^{2+}$ and $Ca_V1.3$ channels also play an important role during inner-ear development. In mammals, prior to hearing onset, auditory hair cells fire spontaneous $Ca^{2+}$ action potentials (*Eckrich et al., 2018*; *Marcotti et al., 2003*; *Tritsch et al., 2007*; *Tritsch et al., 2010*). In mammalian hair cells, these $Ca^{2+}$ action potentials are $Ca_V1.3$-dependent and are thought to be important for synapse and circuit formation. In support of this idea, in vivo work in zebrafish hair cells found that increasing or decreasing voltage-gated $Ca^{2+}$ influx through $Ca_V1.3$ channels during development led to the formation of smaller or larger ribbons respectively (*Sheets et al., 2012*). Furthermore, in mouse knockouts of $Ca_V1.3$, auditory outer hair cells have reduced afferent innervation and synapse number (*Ceriani et al., 2019*). Mechanistically, how $Ca_V1.3$-channel activity regulates ribbon size and innervation, and whether hair-cell $Ca^{2+}$ stores play a role in this process is not known.

Cumulative work has shown that ribbon size varies between species and sensory epithelia (reviewed in *Moser et al., 2006*); these variations are thought to reflect important encoding requirements of a given sensory cell (*Matthews and Fuchs, 2010*). In auditory hair cells, excitotoxic noise damage can also alter ribbon size and lead to hearing deficits (*Jensen et al., 2015*; *Liberman et al., 2015*). Excitotoxic damage is thought to be initiated by mito-$Ca^{2+}$ overload and subsequent ROS production (*Böttger and Schacht, 2013*; *Wang et al., 2018*). Mechanistically, precisely how ribbon size is established during development or altered under pathological conditions is not fully understood.

One known way to regulate ribbon size is through its main structural component Ribeye (*Schmitz et al., 2000*). Perhaps unsurprisingly, previous work has shown that overexpression or depletion of Ribeye in hair cells can increase or decrease ribbon size respectively (*Becker et al., 2018*; *Jean et al., 2018*; *Lv et al., 2016*; *Sheets, 2017*; *Sheets et al., 2011*). Ribeye is a splice variant of the transcriptional co-repressor Carboxyl-terminal binding protein 2 (CtBP2) – a splice variant that is unique to vertebrate evolution (*Schmitz et al., 2000*). Ribeye contains a unique A-domain and a B-domain that is nearly identical to full-length CtBP2. The B-domain contains a nicotinamide

adenine dinucleotide ($NAD^+$, NADH or NAD(H)) binding site (*Magupalli et al., 2008*; *Schmitz et al., 2000*). NAD(H) redox is linked to mitochondrial metabolism (*Srivastava, 2016*). Because CtBPs are able to bind and detect $NAD^+$ and NADH levels, they are thought to function as metabolic biosensors (*Stankiewicz et al., 2014*). For example, previous work has demonstrated that changes in NAD(H) redox can impact CtBP oligomerization and its transcriptional activity (*Fjeld et al., 2003*; *Thio et al., 2004*). Interestingly, in vitro work has shown that both $NAD^+$ and NADH can also promote interactions between Ribeye domains (*Magupalli et al., 2008*). Whether $NAD^+$ or NADH can impact Ribeye interactions and ribbon formation or stability has not been confirmed in vivo.

In neurons, it is well established that during presynaptic activity, mitochondria clear and store $Ca^{2+}$ at the presynapse (*Devine and Kittler, 2018*). Additionally, presynaptic activity and mito-$Ca^{2+}$ can couple together to influence cellular bioenergetics, including NAD(H) redox homeostasis (reviewed in: *Kann and Kovács, 2007*; *Llorente-Folch et al., 2015*). Based on these studies, we hypothesized that $Ca^{2+}$ influx through $Ca_V1.3$ channels may regulate mito-$Ca^{2+}$, which in turn could regulate NAD(H) redox. Changes to cellular bioenergetics and NAD(H) redox could function to control Ribeye interactions and ribbon formation or impact ribbon-synapse function and stability.

To study the impact of mito-$Ca^{2+}$ and NAD(H) redox on ribbon synapses, we examined hair cells in the lateral-line system of larval zebrafish. This system is advantageous because it contains hair cells with easy access for in vivo pharmacology, mechanical stimulation and imaging cellular morphology and function. Within the lateral-line, hair cells are arranged in clusters called neuromasts. The hair cells and ribbon synapses in each cluster form rapidly between 2 to 3 days post-fertilization (dpf) but by 5–6 dpf, the majority of hair cells are mature, and the system is functional (*Kindt et al., 2012*; *McHenry et al., 2009*; *Metcalfe, 1985*; *Murakami et al., 2003*; *Santos et al., 2006*). Thus, these two ages (2–3 dpf and 5–6 dpf) can be used to study mito-$Ca^{2+}$ and NAD(H) redox in developing and mature hair cells respectively.

Using this sensory system, we find that presynaptic-$Ca^{2+}$ influx drives mito-$Ca^{2+}$ uptake. In mature hair cells, mito-$Ca^{2+}$ uptake occurs during evoked stimulation and is required to sustain presynaptic function and ultimately synapse integrity. In developing hair cells, mito-$Ca^{2+}$ uptake coincides with spontaneous rises in presynaptic $Ca^{2+}$. Blocking these spontaneous changes in $Ca^{2+}$ leads to the formation of larger ribbons. Using a redox biosensor, we demonstrate that specifically in developing hair cells, decreasing mito-$Ca^{2+}$ levels increases the $NAD^+$/NADH redox ratio. Furthermore, we show that application of $NAD^+$ or NADH can promote the formation of larger or smaller ribbons respectively. Overall, our results suggest that in hair cells presynaptic-$Ca^{2+}$ influx and mito-$Ca^{2+}$ uptake couple in hair cells to impact ribbon formation and function.

## Results

### Mitochondria are located near presynaptic ribbons

In neurons, synaptic mitochondria have been shown to influence synapse size, plasticity and function (*Flippo and Strack, 2017*; *Todorova and Blokland, 2016*). Based on this work, we hypothesized that mitochondria may impact synapses in hair cells. Therefore, we examined the proximity of mitochondria relative to presynaptic ribbons in zebrafish lateral-line hair cells. We visualized mitochondria and ribbons using transmission electron microscopy (TEM) and in vivo using Airyscan confocal microscopy.

Using TEM, we examined sections that clearly captured ribbons (Example, *Figure 1C*). Near the majority of ribbons (81%) we observed a mitochondrion in close proximity (<1 μm) (*Figure 1D*, median ribbon-to-mitochondria distance = 174 nm, n = 17 out of 21 ribbons). To obtain a more comprehensive understanding of the 3D morphology and location of mitochondria relative to ribbons in live cells, we used Airyscan confocal microscopy. To visualize these structures in living cells, we used transgenic zebrafish expressing MitoGCaMP3 (*Esterberg et al., 2014*) and Ribeye a-tagRFP (*Sheets et al., 2014*) in hair cells to visualize mitochondria and ribbons respectively. Using this approach, we observed tubular networks of mitochondria extending from apex to base (*Figure 1A–B,E–E'*, *Figure 1—figure supplement 1A*, *Video 1*). At the base of the hair cell, we observed ribbons nestled between branches of mitochondria. Overall our TEM and Airyscan imaging suggests that in lateral-line hair cells, mitochondria are present near ribbons.

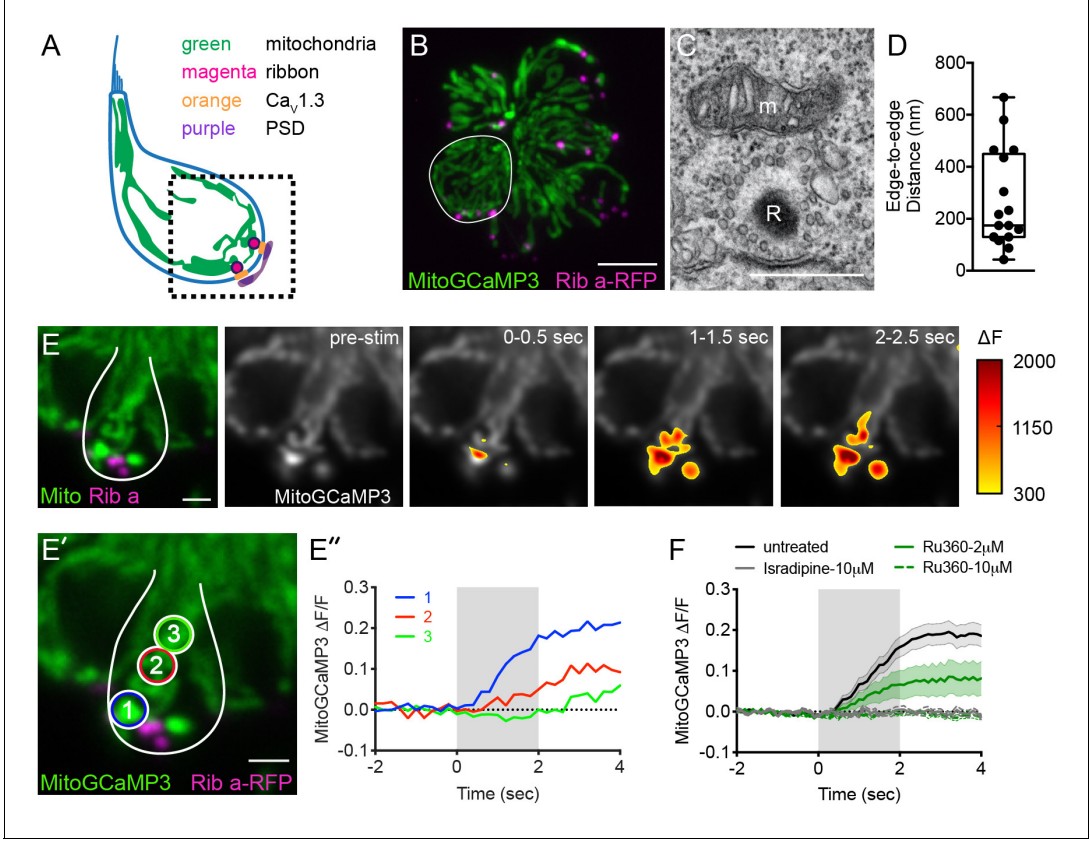

**Figure 1.** Mito-Ca$^{2+}$ uptake initiates adjacent to ribbons. (**A**) Cartoon illustration of a lateral-line hair cell containing: an apical mechanosensory bundle (blue), mitochondria (green), presynaptic ribbons (magenta), Ca$_V$1.3 channels (orange) and postsynaptic densities (purple). (**B**) Airyscan confocal image of 6 live hair cells (1 cell outlined in white) expressing MitoGCaMP3 (mitochondria) and Ribeye a-tagRFP (ribbons) in a developing neuromast at 2 dpf. Also see *Figure 1—figure supplement 1*. (**C**) A representative TEM showing a mitochondrion (m) in close proximity to a ribbon (R) at 4 dpf. (**D**) Quantification of mitochondrion to ribbon distance in TEM sections (n = 17 ribbons). (**E**) Side-view of a hair cell (outlined in white) shows the spatio-temporal dynamics of evoked mito-Ca$^{2+}$ signals during a 2 s stimulation at 6 dpf. The change in MitoGCaMP3 signal (ΔF) from baseline is indicated by the heatmap and are overlaid onto the pre-stimulus grayscale image. (**E'-E''**) Circles 1–3 (1.3 μm diameter) denote regions used to generate the normalized (ΔF/F$_0$) temporal traces of mito-Ca$^{2+}$ signals in E'': adjacent to the presynapse ('1'), and midbody ('2' and '3') in the same cell as E. (**F**) Average evoked mito-Ca$^{2+}$ response before (solid black) and after 30 min treatment with 10 μM Ru360 (dashed green), 2 μM Ru360 (solid green), or 10 μM isradipine (gray) (3–5 dpf, n ≥ 9 cells per treatment). Error bars in D are min and max; in F the shaded area denotes SEM. Scale bar = 500 nm in C, 5 μm in B and 2 μm in E and E'.

The online version of this article includes the following source data and figure supplement(s) for figure 1:

**Source data 1.** Summary of quantified TEM data and mito-Ca$^{2+}$ trace data.

**Figure supplement 1.** The time course of mechanically-evoked mito-Ca$^{2+}$ signals are longer-lasting than cyto-Ca$^{2+}$ signals.

**Figure supplement 1—source data 1.** Summary CytoGCaMP3 and MitoGCaMP3 traces and MitoGCaMP3 data used to generate Ru360 dose response curve.

**Figure supplement 2.** Mito-Ca$^{2+}$ uptake occurs in anterior lateral-line hair cells.

**Figure supplement 2—source data 1.** MitoGCaMP3 traces in anterior lateral line hair cells.

## Mito-Ca$^{2+}$ uptake at ribbons is MCU and Ca$_V$1.3 dependent

In zebrafish hair cells, robust rises in mito-Ca$^{2+}$ have been reported during mechanical stimulation (*Pickett et al., 2018*). Due to the proximity of the mitochondria to the ribbon, we predicted that rises in mito-Ca$^{2+}$ levels during mechanical stimulation could be related to presynapse-associated rises in Ca$^{2+}$.

To test this prediction, we used a fluid-jet to mechanically stimulate hair cells and evoke presynaptic activity. During stimulation, we used MitoGCaMP3 to monitor mito-Ca$^{2+}$ in lateral-line hair

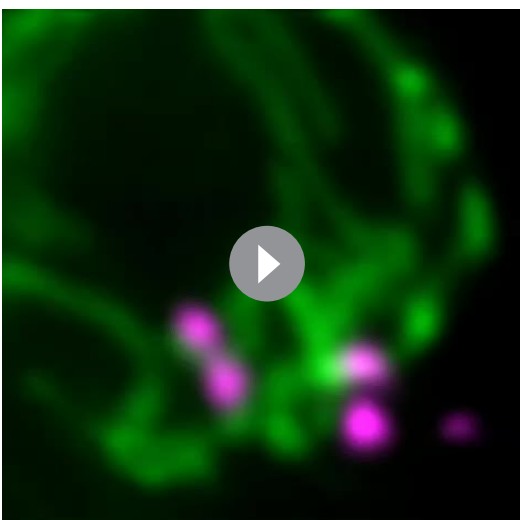

**Video 1.** Airyscan image of MitoGCaMP3 and Rib a-tagRFP at the base of a single live hair cell. https://elifesciences.org/articles/48914#video1

cells. As previously reported, we observed robust mito-Ca$^{2+}$ uptake during stimulation (*Figure 1E–F*, *Figure 1—figure supplement 2*). We examined the subcellular distribution of MitoGCaMP3 signals over time and observed an increase in MitoGCaMP3 ($\Delta F$) signals that initiated near ribbons (*Figure 1E*, $\Delta F$). During the latter part of the stimulus, and even after the stimulus terminated, the MitoGCaMP3 signals propagated apically, away from the ribbons (Example, *Figure 1E'–E''*, regions 1–3, $\Delta F/F_0$). We characterized the time-course of MitoGCaMP3 signals with regards to onset kinetics and return to baseline. During a 2 s stimulus, we detected a significant rise in MitoGCaMP3 signals 0.6 s after stimulus onset (*Figure 1—figure supplement 1B*, $\Delta F/F_0$). Interestingly, after the stimulus terminated, MitoGCaMP3 levels took approximately 5 min to return to baseline (*Figure 1—figure supplement 1C–C'*, $\Delta F/F_0$). Despite this long time-course of recovery to baseline, we were still able to evoke additional rises in MitoGCaMP3 signal 10 s after stimulation (*Figure 1—figure supplement 1D*, $\Delta F/F_0$). As previously reported, the kinetics of MitoGCaMP3 signals in hair-cell mitochondria were quite different from signals observed using cytosolic GCaMP3 (CytoGCaMP3) in hair cells (*Pickett et al., 2018*). Compared to MitoGCaMP3 signals, CytoGCaMP3 signals had faster onset kinetics and a faster return to baseline (*Figure 1—figure supplement 1B–C*, time to rise: 0.06 s, post-stimulus return to baseline: 12 s). These differences in kinetics indicate that mito-Ca$^{2+}$ loading operates over slower timescales compared to the cytosolic compartment. It also confirms that hair-cell stimulation can initiate long lasting increases in mito-Ca$^{2+}$.

To verify that MitoGCaMP3 signals reflect Ca$^{2+}$ entry into mitochondria, we applied Ru360, an antagonist of the mito-Ca$^{2+}$ uniporter (MCU). The MCU is the main pathway for rapid Ca$^{2+}$ entry into the mitochondrial matrix (*Matlib et al., 1998*). We found that stimulus-evoked MitoGCaMP3 signals were blocked in a dose-dependent manner after a 30 min treatment with Ru360 (*Figure 1F*, *Figure 1—figure supplement 1F*; IC$_{50}$ = 1.37 μM). We confirmed these results by applying TRO 19622, an antagonist of the voltage-dependent anion channel (VDAC). VDAC enables transport of ions including Ca$^{2+}$ across the outer mitochondrial membrane (*Schein et al., 1976*; *Shoshan-Barmatz and Gincel, 2003*). We observed that similar to the MCU antagonist Ru360, a 30 min treatment with the VDAC antagonist TRO 19622 also impaired stimulus-evoked MitoGCaMP3 signals (10 μM TRO 19622, *Figure 1—figure supplement 1E*). Due to the initiation of mito-Ca$^{2+}$ near ribbons, we examined whether presynaptic-Ca$^{2+}$ influx through Ca$_V$1.3 channels was the main source of Ca$^{2+}$ entering the mitochondria. To examine Ca$_V$1.3 channel contribution to mito-Ca$^{2+}$ uptake, we applied isradipine, a Ca$_V$1.3 channel antagonist. Similar to blocking the MCU and VDAC, blocking Ca$_V$1.3 channels eliminated all stimulus-evoked MitoGCaMP3 signals at the base of the cell (*Figure 1F*).

Previous work in zebrafish-hair cells demonstrated that isradipine specifically blocks Ca$_V$1.3 channels without impairing mechanotransduction (*Zhang et al., 2018*). For our current study we confirmed whether Ru360 and TRO 19622 specifically block synaptic mito-Ca$^{2+}$ uptake without impairing mechanotransduction. We measured apical, mechanically evoked Ca$^{2+}$ signals in hair bundles before and after a 30 min treatment with 10 μM Ru360 or TRO 19622. Neither compound blocked mechanotransduction (*Figure 2—figure supplement 1A–B'*). Overall our MitoGCaMP3 functional imaging indicates that in hair cells, evoked mito-Ca$^{2+}$ uptake initiates near ribbons and this uptake is dependent on MCU, VDAC and Ca$_V$1.3 channel function.

## Mito-Ca$^{2+}$ uptake occurs in cells with presynaptic-Ca$^{2+}$ influx

Interestingly, we observed that mito-Ca$^{2+}$ uptake was only present in ~40% of cells (Examples, *Figure 2A'* and *Figure 1—figure supplement 2*; n = 10 neuromasts, 146 cells). This observation is

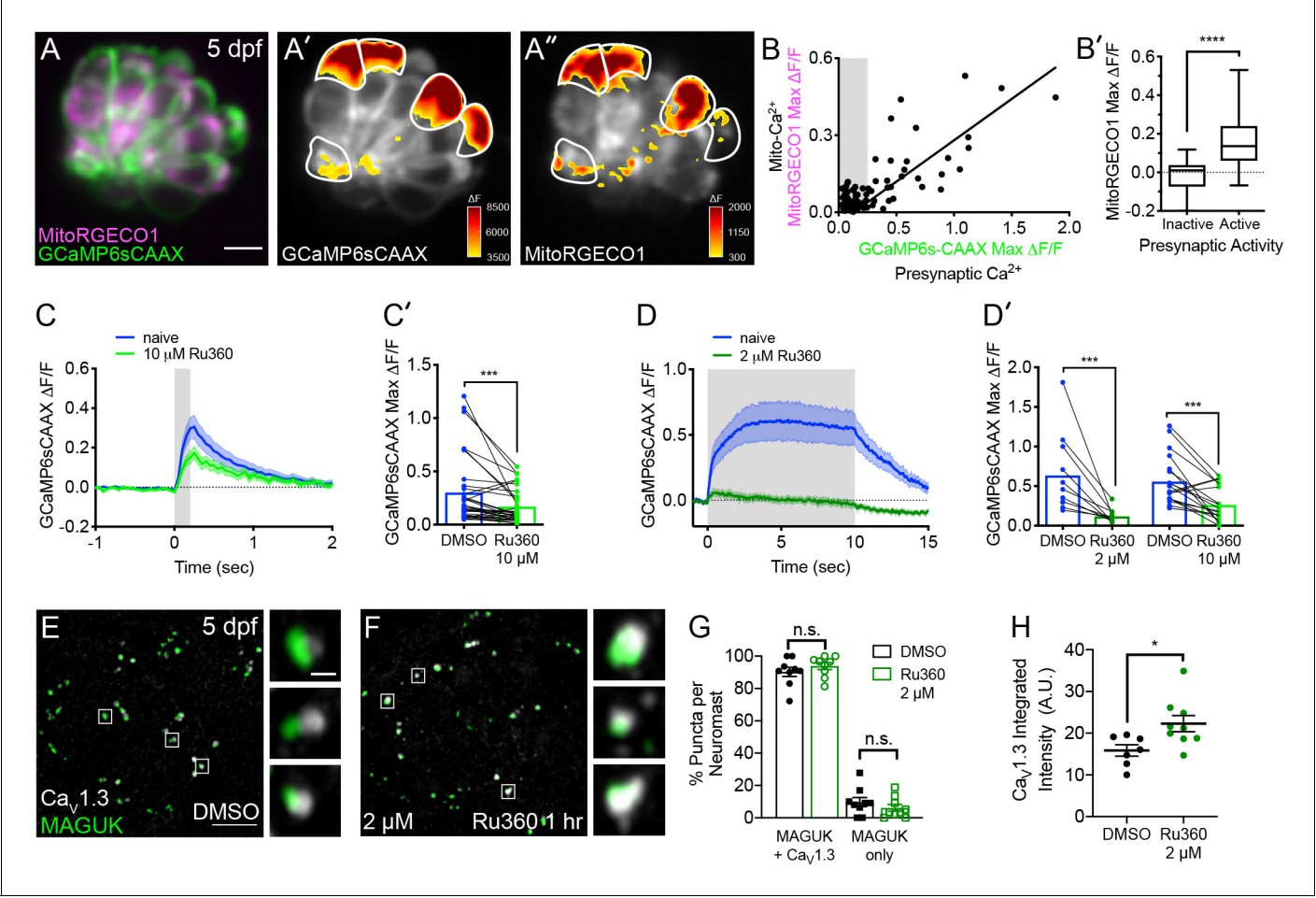

**Figure 2.** Mito-Ca$^{2+}$ uptake can impact presynaptic-Ca$^{2+}$ signals. (A) A live Image of a neuromast viewed top-down, expressing the presynaptic-Ca$^{2+}$ sensor GCaMP6sCAAX (green) and mito-Ca$^{2+}$ sensor MitoRGECO1 (magenta) at 5 dpf. A'-A'', GCaMP6sCAAX (A') and MitoRGECO1 (A'') signals ($\Delta F$) from baseline during a 2 s stimulation are indicated by the heatmaps and occur in the same cells (white outline). (B) Scatter plot with linear regression of peak presynaptic- and mito-Ca$^{2+}$ response for individual cells at 4–5 dpf, n = 136 cells. Gray background in graph denotes presynaptic-Ca$^{2+}$ signals below 0.25, a threshold used as a cutoff for presynaptic activity (below inactive, above active). (B') Plot of mito-Ca$^{2+}$ responses segregated based on the activity threshold in B. (C-D') Presynaptic-Ca$^{2+}$ response (example in *Figure 2—figure supplement 1C–C'*) averaged per cell before (blue) and after a 30 min treatment with 10 µM Ru360 (light green) or 2 µM Ru360 (dark green), n ≥ 10 cells per treatment. C and D show averaged traces while C' and D' show before-and-after dot plots of the peak response per cell. (E-F) Representative images of mature neuromasts (5 dpf) immunostained with Ca$_V$1.3 (white, calcium channels) and MAGUK (green, postsynapses) after a 1 hr incubation in 0.1% DMSO (E) or 2 µM Ru360 (F). G-H, Scatter plots show percentage of postsynapses that pair with Ca$_V$1.3-channel clusters (Ca$_V$1.3 + MAGUK) and orphan postsynapses (MAGUK only) (G). The integrated intensity of Ca$_V$1.3-channel immunolabel at presynapses is lower in control compared to treatment group (H), n ≥ 7 neuromasts per treatment. Whiskers on plots in B' represent min and max; the shaded area in plots C and D and the error bars in C', D' and G-H denotes SEM. Mann-Whitney U test was used in B'; Wilcoxon matched-pairs signed-rank test was used in C' and D'. Welch's unequal variance *t*-test was used in G-H. *p<0.05, ***p<0.001, ****p<0.0001. Scale bar = 5 µm in A and E.

The online version of this article includes the following source data and figure supplement(s) for figure 2:

**Source data 1.** Summary of MitoRGECO1 and GCaMP6sCAAX and synapse quantification data.

**Figure supplement 1.** The effects of MCU and VDAC block on mechanotransduction and the effect of VDAC block on presynaptic-Ca$^{2+}$ signals.

**Figure supplement 1—source data 1.** Mechanosensitive and presynaptic GCaMP6sCAAX traces and quantification after Ru360 and TRO 19622 application.

consistent with previous work demonstrating that only ~30% of hair cells within each neuromast cluster have presynaptic-Ca$^{2+}$ signals and are synaptically active (*Zhang et al., 2018*). Because presynaptic-Ca$^{2+}$ signals initiate near mitochondria, it is probable that mito-Ca$^{2+}$ uptake may occur specifically in hair cells with synaptic activity.

To test whether evoked mito-Ca$^{2+}$ uptake occurred exclusively in cells with presynaptic-Ca$^{2+}$ influx, we performed two-color functional imaging. We used a double transgenic approach that utilized a membrane-localized GCaMP6s (GCaMP6sCAAX; green) to measure presynaptic-Ca$^{2+}$ signals at the base of hair cells (*Jiang et al., 2017*; *Sheets et al., 2017*), and we concurrently used MitoRGECO1 (red) to examine mito-Ca$^{2+}$ signals (*Figure 2A–B'*). Our two-color imaging approach revealed a strong correlation between the magnitude of evoked GCaMP6sCAAX and MitoRGECO1 signals (*Figure 2B*, R$^2$ = 0.77, p<0.0001; n = 136 cells). We found that the median MitoRGECO1 signals were 100% larger in presynaptically active hair cells compared to presynaptically silent hair cells (*Figure 2B'*). Together these results suggest that mito-Ca$^{2+}$ uptake occurs specifically in hair cells with evoked presynaptic-Ca$^{2+}$ influx.

## Blocking mito-Ca$^{2+}$ entry impairs presynaptic-Ca$^{2+}$ signals in mature hair cells

Although we observed mito-Ca$^{2+}$ uptake specifically in hair cells with active Ca$^{2+}$ channels, the impact of mito-Ca$^{2+}$ uptake on the function of hair-cell synapses was unclear. Based on previous studies in neurons and bipolar-cell ribbon synapses (*Billups and Forsythe, 2002*; *Chouhan et al., 2010*; *Kwon et al., 2016*; *Levy et al., 2003*; *Zenisek and Matthews, 2000*), we reasoned that mitochondria may be important to remove excess Ca$^{2+}$ or provide ATP for hair-cell neurotransmission.

To determine if mito-Ca$^{2+}$ uptake impacted presynaptic function, we assayed evoked presynaptic-Ca$^{2+}$ signals by monitoring GCaMP6sCAAX signals adjacent to ribbons as described previously (*Figure 2—figure supplement 1C–C'*; *Sheets et al., 2017*; *Zhang et al., 2018*). We examined GCaMP6sCAAX signals in mature hair cells at 5–6 dpf when neuromast organs are largely mature (*Kindt et al., 2012*; *McHenry et al., 2009*; *Metcalfe, 1985*; *Murakami et al., 2003*; *Santos et al., 2006*). Using this approach, we assayed presynaptic GCaMP6sCAAX signals before and after a 30 min application of the MCU antagonist Ru360 (*Figure 2C–D'*). We found that during short, 200 ms stimuli, GCaMP6sCAAX signals at ribbons were reduced after complete MCU block (10 µM Ru360, *Figure 2C–C'*). Reduction of GCaMP6sCAAX signals were further exacerbated during sustained 10 s stimuli, even when the MCU was only partially blocked (2 µM Ru360, *Figure 2D–D'*). A similar reduction in GCaMP6sCAAX signals were observed after a 30 min application of the VDAC inhibitor TRO 19622 (*Figure 2—figure supplement 1D-E'*, 10 µM TRO 19622). These results suggest that in mature hair cells, evoked mito-Ca$^{2+}$ uptake is critical for presynaptic-Ca$^{2+}$ influx, especially during sustained stimulation.

## Evoked mito-Ca$^{2+}$ uptake is important for mature synapse integrity and cell health

MCU block could impair presynaptic-Ca$^{2+}$ influx through several mechanisms. It could impair the biophysical properties of Ca$_V$1.3 channels, for example, through Ca$^{2+}$-dependent inactivation (*Platzer et al., 2000*; *Schnee and Ricci, 2003*). MCU block could also impact Ca$_V$1.3 channel localization. In addition, mito-Ca$^{2+}$ has been implicated in synapse dysfunction and cell death (*Esterberg et al., 2014*; *Vos et al., 2010*; *Wang et al., 2018*), and MCU block could be pathological. To distinguish between these possibilities, we assessed whether synaptic components or hair-cell numbers were altered after MCU block with Ru360.

To quantify ribbon-synapse morphology after MCU block, we immunostained mature-hair cells (5 dpf) with Ca$_V$1.3, Ribeye b and MAGUK antibodies to label Ca$_V$1.3 channels, presynaptic ribbons and postsynaptic densities (MAGUK) respectively. We first applied 2 µM Ru360 for 1 hr, a concentration that partially reduces evoked mito-Ca$^{2+}$ uptake (See *Figure 1F*), yet is effective at reducing sustained presynaptic-Ca$^{2+}$ influx (See *Figure 2D–D'*). At this dose, Ru360 had no impact on hair cell or synapse number (*Figure 3E*). We also observed no morphological change in ribbon or postsynapse size (*Figure 3F*, *Figure 3—figure supplement 1C*, *Figure 3—figure supplement 2*). After the 1 hr 2 µM Ru360 treatment, Ca$_V$1.3 clusters were still present at synapses, but the channels were at a significantly higher density compared to controls (*Figure 2E–H*). These findings indicate that in mature hair cells, partial MCU block may impair presynaptic function by altering Ca$_V$1.3 channel density.

We also tested a higher dose of Ru360 (10 µM) that completely blocks evoked mito-Ca$^{2+}$ uptake (See *Figure 1F*). Interestingly, a 30 min or 1 hr 10 µM Ru360 treatment had a progressive impact on synapse and cellular integrity. After a 30 min treatment with 10 µM Ru360 we did not observe fewer

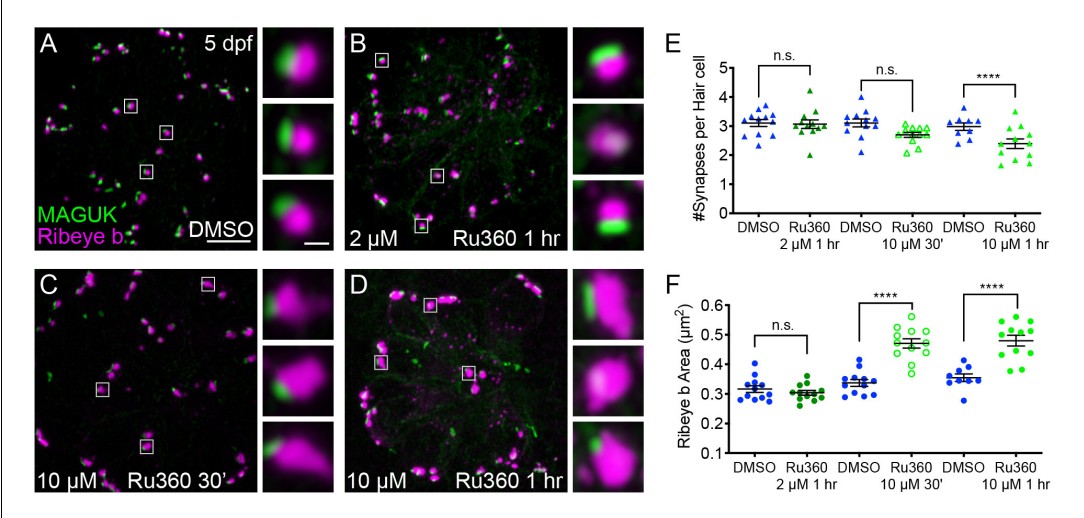

**Figure 3.** Mito-Ca$^{2+}$ is important for ribbon size and synapse integrity in mature hair cells. (A-D) Representative images of mature neuromasts (5 dpf) immunostained with Ribeye b (magenta, ribbons) and MAGUK (green, postsynapses) after a 1 hr 0.1% DMSO (A), a 1 hr 2 µM Ru360 (B), a 30 min 10 µM Ru360 (C), or a 1 hr 10 µM Ru360 (D) treatment. Insets show three example synapses (white squares). E-F, Scatter plots show synapse counts (E), and ribbon area (F) in controls and in treatment groups. Ribbon areas, synapse numbers, and hair-cell counts are unaffected after a 1 hr 2 µM Ru360 treatment. Ribbon areas are larger and there are fewer synapses without significant loss of hair cells after a 30 min treatment with 10 µM Ru360 (F). After a 1 hr 10 µM Ru360 treatment there is an increase in ribbon area and a decrease in synapse (E) and hair-cell counts. N ≥ 9 neuromasts per treatment. Error bars in E-F represent SEM. An unpaired *t*-test was used in E and a Welch's unequal variance *t*-test was used in F. ****p<0.0001. Scale bar = 5 µm in A, and 2 µm in inset.

The online version of this article includes the following source data and figure supplement(s) for figure 3:

**Source data 1.** Summary of synapse number and ribbon area after Ru360.
**Figure supplement 1.** Ribbon and postsynapse size in mature ALL neuromasts.
**Figure supplement 1—source data 1.** Summary of data comparing anterior and posterior lateral-line synapses in mature hair cells.
**Figure supplement 2.** MCU block does not impact postsynapse size in mature hair cells.
**Figure supplement 2—source data 1.** Summary of MAGUK area measurements after Ru360 treatments in mature hair cells.

complete synapses per hair cell or fewer hair cells compared to controls (*Figure 3E*; Hair cells per neuromast, control: 16.3, 30 min 10 µM Ru360: 15.5; p=0.5). But after the 30 min treatment, ribbons were significantly larger (*Figure 3F*). The effects of MCU block became more pathological after a 1 hr, 10 µM Ru360 treatment. After 1 hr, there were both fewer hair cells per neuromast (Hair cells per neuromast, control: 18.1, 1 hr 10 µM Ru360: 12.0; p>0.0001) and fewer synapses per hair cell (*Figure 3E*). Similar to 30 min treatments with Ru360, after 1 hr, ribbons were also significantly larger (*Figure 3F*). Neither 30 min nor 1 hr 10 µM Ru360 treatment altered postsynapse size (*Figure 3—figure supplement 2*). Overall, our results indicate that in mature hair cells, partial block of mito-Ca$^{2+}$ uptake may impair presynaptic function by altering Ca$_V$1.3 channel clustering, without seemingly altering other gross pre- or post-synaptic morphology. Complete block of mito-Ca$^{2+}$ uptake is pathological; it impairs presynaptic function, alters presynaptic morphology, and results in a loss of synapses and hair-cells.

## Spontaneous presynaptic and mito-Ca$^{2+}$ influx pair in developing hair cells

In addition to evoked presynaptic- and mito-Ca$^{2+}$ signals in hair cells, we also observed instances of spontaneous presynaptic- and mito-Ca$^{2+}$ signals (Example, *Figure 4A–A'''*, *Video 2*). Numerous studies have demonstrated that mammalian hair cells have spontaneous presynaptic-Ca$^{2+}$ influx during development (*Eckrich et al., 2018*; *Holman et al., 2019*; *Marcotti et al., 2003*; *Tritsch et al., 2007*; *Tritsch et al., 2010*). Therefore, we predicted that similar to mammals, spontaneous presynaptic-Ca$^{2+}$ uptake may be a feature of development. Furthermore, we predicted that spontaneous mito-Ca$^{2+}$ uptake may correlate with instances of spontaneous presynaptic-Ca$^{2+}$ influx.

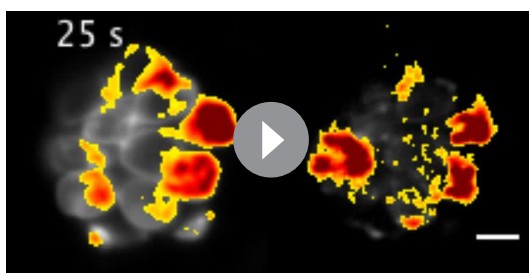

**Video 2.** Spontaneous ΔF GCaMP6sCAAX (left) and ΔF MitoRGECO1 (right) signals acquired at 3 dpf, 25 s per frame.
https://elifesciences.org/articles/48914#video2

First, we tested whether spontaneous presynaptic-$Ca^{2+}$ signals were a feature of development. In zebrafish neuromasts, hair cells are rapidly added between 2–3 dpf, but by 5–6 dpf relatively fewer cells are added and the hair cells and the organs are largely mature (*Kindt et al., 2012*; *McHenry et al., 2009*; *Metcalfe, 1985*; *Murakami et al., 2003*; *Santos et al., 2006*). Therefore, we examined the magnitude and frequency of spontaneous, presynaptic GCaMP6sCAAX signals in developing (3 dpf) and mature (5 dpf) hair cells. We found that in developing hair cells, spontaneous GCaMP6sCAAX signals occurred with larger magnitudes and at a higher frequency compared to those in mature hair cells (*Figure 4B–C*). Our spontaneous GCaMP6sCAAX imaging demonstrates that similar to mammals, spontaneous presynaptic-$Ca^{2+}$ activity is a feature of developing zebrafish hair cells.

Next we tested whether spontaneous mito-$Ca^{2+}$ uptake and presynaptic-$Ca^{2+}$ influx were correlated. For this analysis we concurrently imaged GCaMP6sCAAX and MitoRGECO1 signals in the same cells for 15 mins to measure presynaptic- and mito-$Ca^{2+}$ responses respectively. We found that spontaneous presynaptic-$Ca^{2+}$ influx was often associated with spontaneous mito-$Ca^{2+}$ uptake (Example, *Figure 4A–A'''*). Overall, we observed a high correlation between the rise and fall of these two signals within individual cells (*Figure 4A''–A'''*). Both of these signals and their correlation were abolished by application of the $Ca_V1.3$-channel antagonist isradipine (*Figure 4—figure supplement 1*). Together these experiments indicate that, similar to our evoked experiments, spontaneous presynaptic- and mito-$Ca^{2+}$ signals are correlated.

## Spontaneous mito-$Ca^{2+}$ uptake regulates ribbon formation

Previous work in zebrafish demonstrated that $Ca_V1.3$ channel activity plays a role in regulating ribbon size specifically during development (*Sheets et al., 2012*). This work found that a transient, 1 hr pharmacological block of $Ca_V1.3$ channels increased ribbon size, while $Ca_V1.3$ channel agonists decreased ribbon size (*Figure 5E*; *Sheets et al., 2012*). Therefore, we reasoned that spontaneous $Ca_V1.3$ and mito-$Ca^{2+}$ activities could function together to control ribbon size in developing hair cells.

To characterize the role of spontaneous mito-$Ca^{2+}$ uptake on ribbon size, we applied the MCU antagonist Ru360 to developing hair cells (3 dpf). After this treatment, we quantified ribbon-synapse morphology by immunostaining hair cells to label presynaptic ribbons and postsynaptic densities. After a 1 hr application of 2 µM Ru360 to block the MCU, we observed a significant increase in ribbon size in developing hair cells (*Figure 5A–B,E*, *Figure 5—figure supplement 1C*). In contrast, this same treatment did not impact ribbon size in mature hair cells (*Figure 3F*, *Figure 3—figure supplement 1C*). We also applied a higher concentration of Ru360 (10 µM) to developing hair cells for 1 hr. In developing hair cells, after a 1 hr 10 µM Ru360 treatment, we also observed a significant increase in ribbon size (*Figure 5A,C,E*). Unlike in mature hair cells (*Figure 3*), in developing hair cells, these concentrations of the MCU antagonist did not alter the number of hair cells or the number of synapses per hair cell (*Figure 5D*; Hair cells per neuromast, control: 9.0, 1 hr 10 µM Ru360: 8.8, p=0.3). All morphological changes were restricted to the ribbons, as MCU block did not alter the size of the postsynapse (*Figure 5—figure supplement 2*).

In addition to larger ribbons, at higher concentrations of Ru360 (10 µM) we also observed an increase in cytoplasmic, non-synaptic Ribeye aggregates (*Figure 5F,G*). Previous work in zebrafish reported both larger ribbons and cytoplasmic aggregates of Ribeye in $Ca_V1.3a$-deficient hair cells (*Sheets et al., 2011*). These parallel phenotypes indicate that spontaneous presynaptic-$Ca^{2+}$ influx and mito-$Ca^{2+}$ uptake may couple to shape Ribeye aggregation and ribbon size. Our results suggest that during development, spontaneous $Ca^{2+}$ entry through both $Ca_V1.3$ and MCU channels continuously regulate ribbon formation; blocking either channel increases Ribeye aggregation and ribbon size.

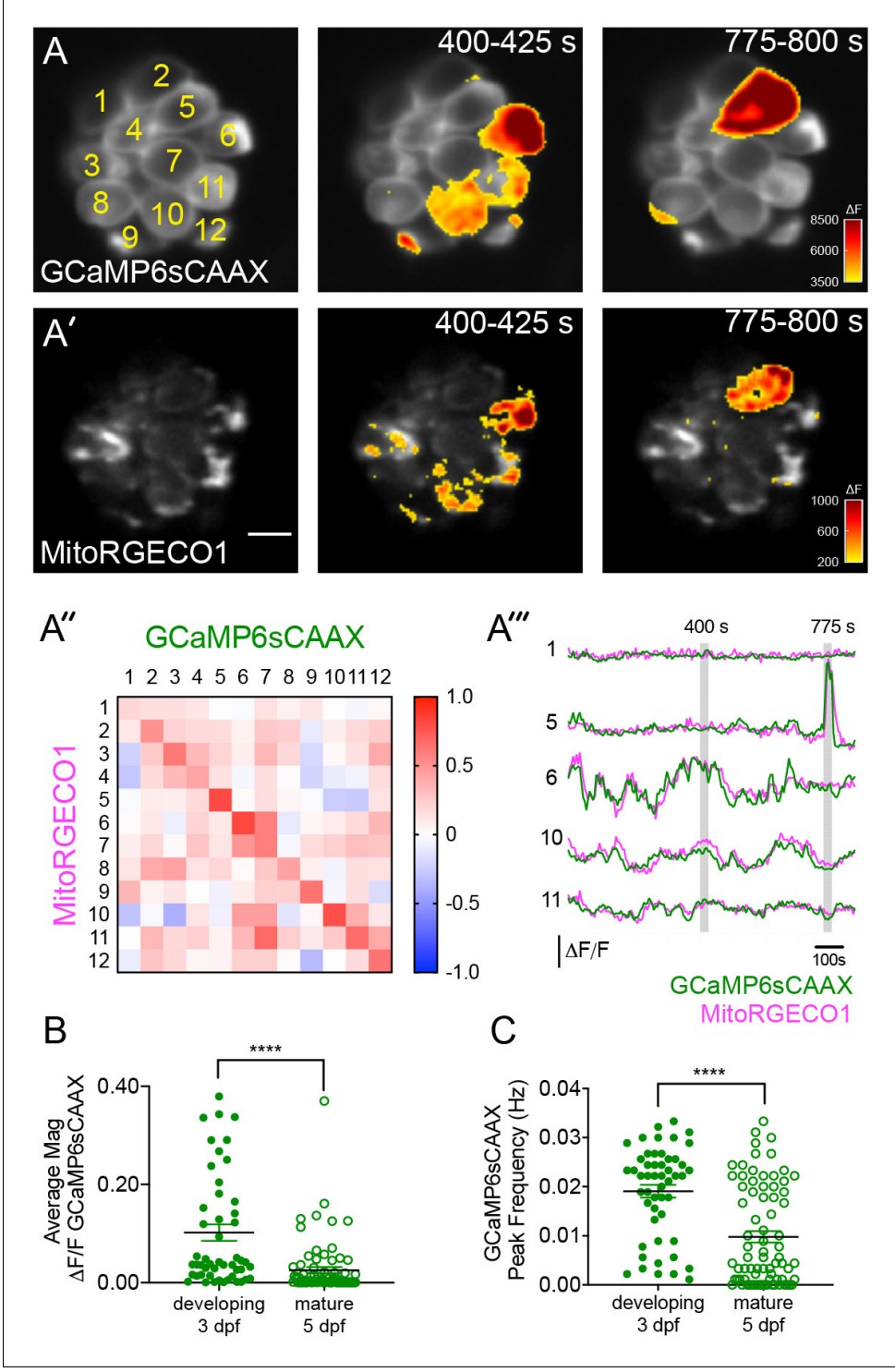

**Figure 4.** Spontaneous presynaptic- Ca²⁺ influx and mito-Ca²⁺ uptake are linked. (A-A') A live Image of an immature neuromast viewed top-down, expressing the presynaptic-Ca²⁺ sensor GCaMP6sCAAX (A) and mito-Ca²⁺ sensor MitoRGECO1 (A') at 3 dpf. Example GCaMP6sCAAX (A') and MitoRGECO1 (A') signals during two 25 s windows within a 900 s acquisition are indicated by the ΔF heatmaps and occur in the same cells. (A'') A heatmap of Pearson correlation coefficients comparing GCaMP6sCAAX and MitoRGECO1 signals from the cells in A-A'. (A''') Example GCaMP6sCAAX (green) MitoRGECO1 (magenta) traces during the 900 s acquisition from the 5 cells numbered in A, also see *Video 2*. (B) Scatter plot showing the average magnitude of GCaMP6sCAAX signals in developing and mature hair cells, n = 6 neuromasts per age. (C) Scatter plot showing frequency of

*Figure 4 continued on next page*

*Figure 4 continued*

GCaMP6sCAAX events in developing and mature hair cells, n = 6 neuromasts. Error bars in B-C represent SEM. A Mann-Whitney U test was used in B and C. ****p<0.0001. Scale bar = 5 µm in A'.

The online version of this article includes the following source data and figure supplement(s) for figure 4:

**Source data 1.** Summary of the magnitude and frequency of spontaneous GCaMP6s-CAAX signals.
**Figure supplement 1.** Spontaneous presynaptic and mito-Ca$^{2+}$ signals are abolished by Ca$_V$1.3 channel antagonist isradipine.
**Figure supplement 1—source data 1.** Summary of MitoRGECO and GCaMP6s traces used to generate correlation plot.

## MCU and Ca$_V$1.3 channel activities regulate subcellular Ca$^{2+}$ homeostasis

Our results indicate that spontaneous Ca$^{2+}$ influx through Ca$_V$1.3 channels and subsequent loading of Ca$^{2+}$ into mitochondria regulates ribbon size in developing hair cells. But how do these two Ca$^{2+}$ signals converge to regulate ribbon size? It is possible that mitochondria could buffer Ca$^{2+}$ during spontaneous presynaptic activity and function to decrease resting levels of cytosolic Ca$^{2+}$ (cyto-Ca$^{2+}$); cyto-Ca$^{2+}$ levels could be a signal that regulates ribbon size. To examine resting cyto-Ca$^{2+}$ levels in hair cells, we examined the fluorescence signal change of the cytosolic Ca$^{2+}$ indicator RGECO1 (CytoRGECO1) before and after a 30 min pharmacological manipulation of Ca$_V$1.3 or MCU channels (*Figure 6A–C*).

We observed that treatment with the Ca$_V$1.3 channel antagonist isradipine and agonist Bay K8644 decreased and increased resting CytoRGECO1 fluorescence respectively (*Figure 6B*).

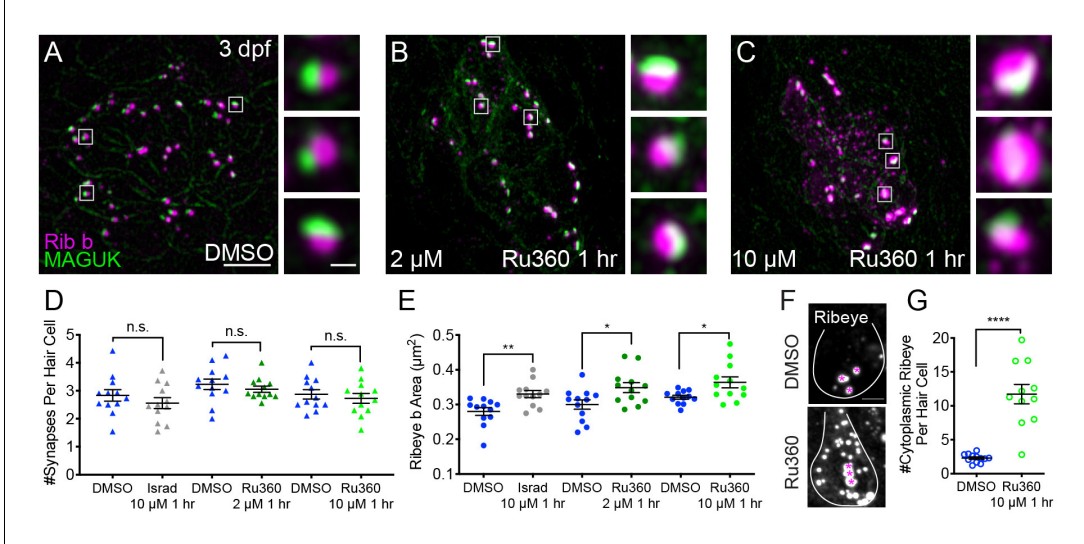

**Figure 5.** Mito-Ca$^{2+}$ regulates ribbon formation. (A-C) Representative images of immature neuromasts (3 dpf) immunostained with Ribeye b (magenta, ribbons) and MAGUK (green, postsynapses) after a 1 hr 0.1% DMSO (A), 2 µM Ru360 (B) or 10 µM Ru360 (C) treatment. Insets show three representative synapses (white squares) for each treatment. D-E, Scatter plot show quantification of synapse number (D), and ribbon area (E) in controls and in treatment groups. (F) Side-view of 2 hair cells (white outline) shows synaptic ribbon (three magenta asterisks in each cell) and extrasynaptic Ribeye b aggregates after a 1 hr 0.1% DMSO or 10 µM Ru360 treatment. (G) Quantification of extrasynaptic Ribeye puncta. N ≥ 12 neuromasts per treatment. Error bars in D-E and G represent SEM. An unpaired *t*-test was used in D and a Welch's unequal variance *t*-test was used in E and G, *p<0.05, **p<0.01, ****p<0.0001. Scale bar = 5 µm in A, 2 µm in insets and F.

The online version of this article includes the following source data and figure supplement(s) for figure 5:

**Source data 1.** Summary of synapse number and ribbon area after Ru360 application in developing hair cells.
**Figure supplement 1.** Ribbon and postsynapse size in immature ALL neuromasts.
**Figure supplement 1—source data 1.** Summary of data comparing anterior and posterior lateral-line synapses in developing hair cells.
**Figure supplement 2.** MCU and Ca$_V$1.3 block do not impact postsynapse size.
**Figure supplement 2—source data 1.** Summary of MAGUK area measurements after Ru360 treatment in developing hair cells.

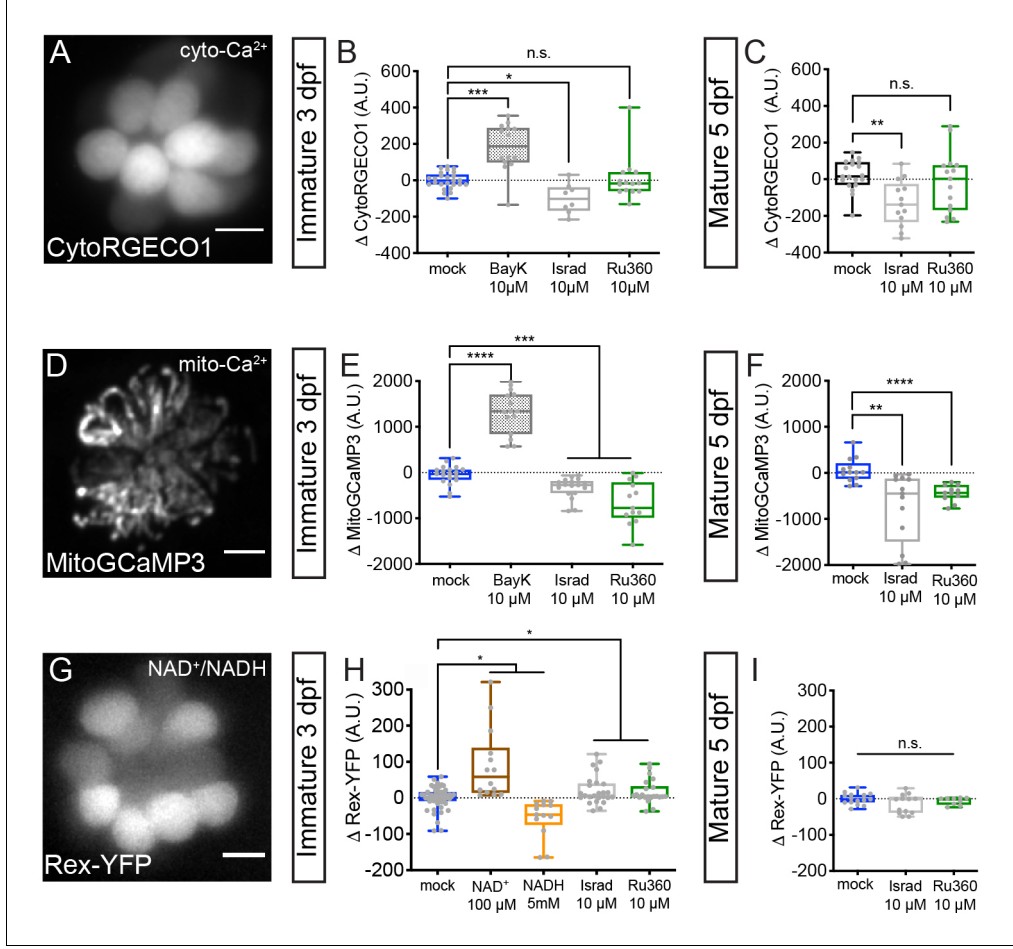

**Figure 6.** Cyto-Ca$^{2+}$, mito-Ca$^{2+}$ and NAD$^+$/NADH redox baseline measurements. Live hair cells expressing RGECO1 (A), MitoGCaMP3 (D), or Rex-YFP (G) show resting cyto-Ca$^{2+}$, mito-Ca$^{2+}$ or NAD$^+$/NADH levels respectively. (B-C) RGECO1 baseline measurements before and after a 30 min mock treatment (0.1% DMSO) or after a 30 min 10 µM Bay K8644 (BayK), 10 µM isradipine, or 10 µM Ru360 treatment. (E-F) MitoGCaMP3 baseline measurements before and after a 30 min mock treatment (0.1% DMSO) or after a 10 µM BayK, 10 µM isradipine, or 10 µM Ru360 treatment. (H-I) Rex-YFP baseline measurements before and after 30 min mock treatment (0.1% DMSO) or after a 30 min 100 µM NAD$^+$, 5 mM NADH, 10 µM isradipine, or 10 µM Ru360 treatment. All plots are box-and-whiskers plot that show median, min and max. N $\geq$ 9 neuromasts per treatment. A one-way Brown-Forsythe ANOVA with Dunnett's T3 post hoc was used to calculate the difference in (B-C), (E-F), and a one-way Brown-Forsythe and Welch ANOVA with Holm-Sidak's post hoc was used in H-I, *p<0.05, **p<0.01, ***p<0.001, ****p<0.0001. Horizontal lines in E, H, and I indicate that both conditions had similar p values compared to mock treatment. Scale bar = 5 µm in A, D and G.

The online version of this article includes the following source data for figure 6:

**Source data 1.** Summary of baseline CytoRGECO1, MitoGCaMP3 and Rex-YFP measurements.

However, treatment with MCU blocker Ru360 did not significantly shift resting CytoRGECO1 fluorescence levels (*Figure 6B*). Similar results with Ru360 were observed in developing and mature hair cells (*Figure 6B–C*). These data suggest that, unlike Ca$_V$1.3 channel function, MCU function and associated mito-Ca$^{2+}$ uptake does not play a critical role in buffering steady state cyto-Ca$^{2+}$ levels.

Alternatively, it is possible that rather than impacting cyto-Ca$^{2+}$ levels, both Ca$_V$1.3 and MCU activity are required to load and maintain Ca$^{2+}$ levels within the mitochondria. In this scenario, mito-Ca$^{2+}$ levels could be a signal that regulates ribbon size. To test this possibility, we used MitoG-CaMP3 to examine resting mito-Ca$^{2+}$ levels before and after modulating Ca$_V$1.3 or MCU channel function (*Figure 6D–F*). We observed that blocking Ca$_V$1.3 channels with isradipine or the MCU with Ru360 decreased resting MitoGCaMP3 fluorescence (*Figure 6E–F*). Conversely, Ca$_V$1.3 channel

agonist Bay K8644 increased resting MitoGCaMP3 fluorescence (*Figure 6E*). These results were consistent in developing and mature hair cells (*Figure 6E–F*). Our resting MitoGCaMP3 measurements indicate that the effects of $Ca_V1.3$ channel and MCU activity converge to regulate mito-$Ca^{2+}$ levels. When either of these channels are blocked, the resting levels of mito-$Ca^{2+}$ decrease. Therefore, if presynaptic-$Ca^{2+}$ influx and mito-$Ca^{2+}$ regulate ribbon size through a similar mechanism, they may act through mito- rather than cyto-$Ca^{2+}$ homeostasis.

## Mito-$Ca^{2+}$ levels regulate NAD(H) redox in developing hair cells

If mito-$Ca^{2+}$ levels signal to regulate ribbon size, how is this signal transmitted from the mitochondria to the ribbon? An ideal candidate is NAD(H) homeostasis. Ribeye protein, the main component of ribbons contains a putative NAD(H) binding site. Because mitochondria regulate NAD(H) redox homeostasis (*Jensen-Smith et al., 2012*), we reasoned that there may be a relationship between mito-$Ca^{2+}$ levels, NAD(H) redox, and ribbon size.

To examine NAD(H) redox, we created a stable transgenic line expressing Rex-YFP, a fluorescent $NAD^+$/NADH ratio biosensor in hair cells (*Figure 6G*). We verified the function of the Rex-YFP biosensor in our in vivo system by exogenously applying $NAD^+$ or NADH for 30 min. We found that incubation with 100 µM $NAD^+$ increased while 5 mM NADH decreased Rex-YFP fluorescence; these intensity changes are consistent with an increase and decrease in the $NAD^+$/NADH ratio respectively (*Figure 6H*). Next, we examined if $Ca_V1.3$ and MCU channel activities impact the $NAD^+$/NADH ratio. We found that 30 min treatments with either a $Ca_V1.3$ or MCU channel antagonist increased the $NAD^+$/NADH ratio (increased Rex-YFP fluorescence) in developing hair cells (*Figure 6H*). Interestingly, similar 30 min treatments did not alter Rex-YFP fluorescence in mature hair cells (*Figure 6I*). Together, our baseline MitoGCaMP3 and Rex-YFP measurements indicate that during development, $Ca_V1.3$ and MCU channel activities normally function to increase mito-$Ca^{2+}$ and decrease the $NAD^+$/NADH ratio. Overall, this work provides strong evidence that links NAD(H) redox and mito-$Ca^{2+}$ with ribbon formation.

## $NAD^+$ and NADH directly influence ribbon size

Our Rex-YFP measurements suggest that in developing hair cells, $Ca_V1.3$ and MCU $Ca^{2+}$ activities normally function to decrease the $NAD^+$/NADH ratio; furthermore, these activities may function to restrict ribbon size. Conversely, blocking these activities increases the $NAD^+$/NADH ratio and may increase ribbon size. If the $NAD^+$/NADH ratio is an intermediate step between $Ca_V1.3$ and MCU channel activities and ribbon formation, we predicted that more $NAD^+$ or NADH would increase or decrease ribbon size respectively. To test this prediction, we treated developing hair cells with exogenous $NAD^+$ or NADH.

After a 1 hr treatment with 100 µM $NAD^+$, we found that the ribbons in developing hair cells were significantly larger compared to controls (*Figure 7A–B,E*). In contrast, after a 1 hr treatment with 5 mM NADH, ribbons were significantly smaller compared to controls (*Figure 7A,C,E*). Neither exogenous $NAD^+$ nor NADH were able to alter ribbon size in mature hair cells (*Figure 7F–H,J*). These concentrations of $NAD^+$ and NADH altered neither the number of synapses per hair cell nor postsynapse size in developing or mature hair cells (*Figure 7D,I*, *Figure 7—figure supplement 1*). These results suggest that in developing hair cells, $NAD^+$ promotes while NADH inhibits Ribeye-Ribeye interactions or Ribeye localization to the ribbon. Overall these results support the idea that during development, the levels of $NAD^+$ and NADH can directly regulate ribbon size in vivo.

## Discussion

In this study, we determined in a physiological setting how mito-$Ca^{2+}$ influences hair-cell presynapse function and formation. In mature hair cells, evoked $Ca_V1.3$-channel $Ca^{2+}$ influx drives $Ca^{2+}$ into mitochondria. Evoked mito-$Ca^{2+}$ uptake is important to sustain presynaptic-$Ca^{2+}$ responses and maintain synapse integrity (*Figure 8B*). During development, spontaneous $Ca_V1.3$ channel $Ca^{2+}$ influx also drives $Ca^{2+}$ into mitochondria. Elevated mito-$Ca^{2+}$ levels rapidly lower the $NAD^+$/NADH ratio and downregulate ribbon size (*Figure 8A*). Furthermore, during development, $NAD^+$ and NADH can directly increase and decrease ribbon size respectively. Our study reveals an intriguing mechanism that couples presynaptic activity with mito-$Ca^{2+}$ to regulate the function and formation of a presynaptic structure.

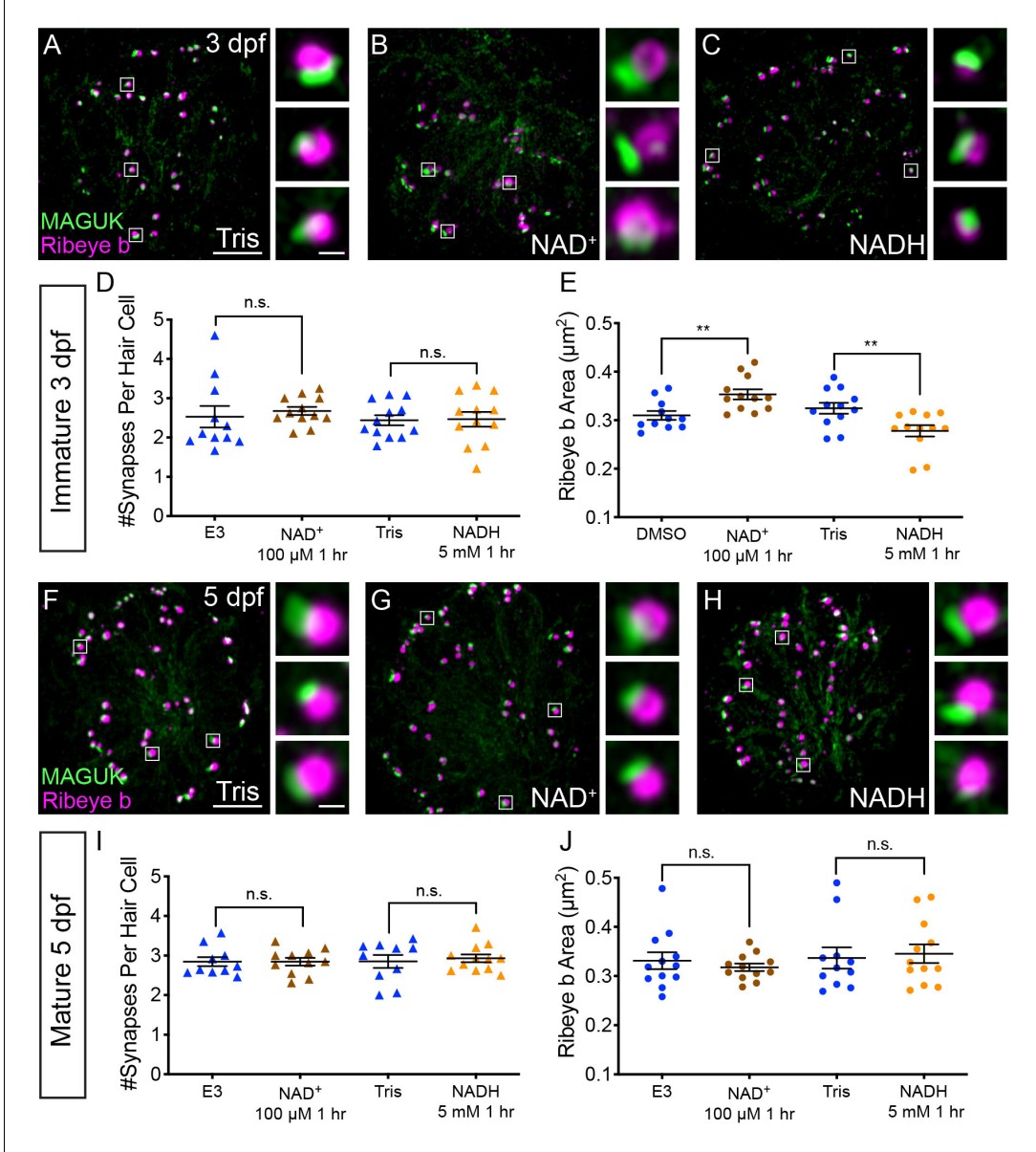

**Figure 7.** NAD+ and NADH directly influence ribbon formation. Representative images of immature (**A-C**, 3 dpf) and mature (**G-H**, 5 dpf) neuromasts immunostained with Ribeye b (magenta, ribbons) and MAGUK (green, postsynapses) after a 0.1% Tris-HCl (**A, F**), 100 μM NAD+ (**B, G**) or 5 mM NADH treatment (**C, H**). Insets show three example synapses (white squares). D-E and I-J, Scatter plots show synapse count (**D, I**) and ribbon area (**E, J**) in controls and treatments groups. N ≥ 10 neuromasts per treatment. Error bars in B-C represent SEM. An unpaired *t*-test was used for comparisons in D and I and a Welch's unequal variance *t*-test was used for comparisons in E and J, **p<0.01. Scale bar = 5 μm in A and F, 2 μm in insets.

The online version of this article includes the following source data and figure supplement(s) for figure 7:

**Source data 1.** Summary of synapse number and ribbon area measurements after NAD+ and NADH application.

**Figure supplement 1.** NAD+ and NADH treatment do not impact postsynapse size.

**Figure supplement 1—source data 1.** Summary of MAGUK area after NAD+ and NADH treatment.

## Functional significance of ribbon size

Our work outlines how during development, presynaptic activity controls the size of ribbons. When either presynaptic-Ca$^{2+}$ influx or mito-Ca$^{2+}$ uptake was perturbed, ribbons were significantly larger (*Figure 5A–C,E*; *Sheets et al., 2012*). But why regulate ribbon size?

Previous work has reported variations in ribbon size and shape among hair-cell types and species (*Moser et al., 2006*). In many instances ribbon size is correlated with functional properties of the

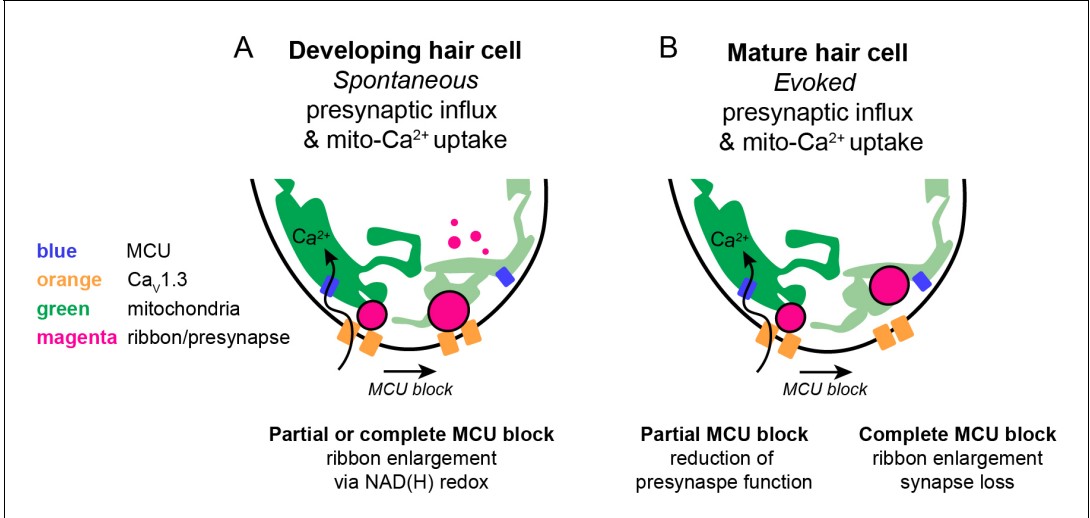

**Figure 8.** Schematic model of mito-Ca$^{2+}$ in developing and mature hair cells. (**A**) In developing hair cells, spontaneous presynaptic-Ca$^{2+}$ influx is linked to mito-Ca$^{2+}$ uptake. Together these Ca$^{2+}$ signals function to regulate ribbon size during ribbon formation. When the Ca$_V$1.3 or MCU channels are blocked, ribbon formation is increased leading to larger ribbons. These Ca$^{2+}$ signals regulate ribbon formation via NAD(H) redox. MCU block lowers mito-Ca$^{2+}$, increases the NAD$^+$/NADH ratio and promotes ribbon formation. (**B**) In mature hair cells, evoked presynaptic-Ca$^{2+}$ influx is linked to mito-Ca$^{2+}$ uptake. When the MCU is partially blocked there is a reduction in presynaptic-Ca$^{2+}$ influx. When the MCU is completely blocked there are synaptopathic consequences; ribbons are enlarged and synapses are lost.

synapse. For example, in the mammalian vestibular system, the ribbons of type II dimorphic hair cells in the striolar region are larger than those in the extrastriolar region (*Lysakowski and Goldberg, 1997*). Functionally, afferents that innervate hair cells with larger ribbons in the striolar region have lower rates of spontaneous activity compared to afferents that innervate hair cells in the extrastriolar region (*Eatock et al., 2008*; *Goldberg et al., 1984*; *Risner and Holt, 2006*). Similarly, in the mammalian auditory system, ribbon size is correlated with differences in afferent activity. Inner hair cells are populated by ribbons with a range of sizes, each of which is innervated by a unique afferent fiber. Compared to smaller ribbons, larger ribbons within inner hair cells are innervated by afferent fibers with higher thresholds of activation and lower rates of spontaneous activity (*Furman et al., 2013*; *Kalluri and Monges-Hernandez, 2017*; *Liberman et al., 2011*; *Liberman et al., 2015*; *Liberman et al., 1990*; *Merchan-Perez and Liberman, 1996*; *Song et al., 2016*; *Yin et al., 2014*). Interestingly, in mice differences in ribbon size can be distinguished just after the onset of hearing (*Liberman and Liberman, 2016*). This timing suggests that similar to our data (*Figures 4–5*), activity during development may help determine ribbon size.

Previous work in the zebrafish-lateral line has also examined how ribbon enlargement impacts synapse function (*Sheets et al., 2017*). This work overexpressed Ribeye in zebrafish hair cells to dramatically enlarge ribbons. Functionally, compared to controls, hair cells with enlarged ribbons were associated with afferent neurons with lower spontaneous activity (*Sheets et al., 2017*). Furthermore, the onset encoding, or the timing of the first afferent spike upon stimulation, was significantly delayed in hair cells with enlarged ribbons. Together, both studies in zebrafish and mammals indicate that ribbon size can impact the functional properties of the synapse. Based on these studies, we predict that the alterations to ribbon size we observed in our current study would impact functional properties of the synapse in a similar manner. For example, pharmacological treatments that enlarge ribbons (*Figure 5*: MCU channel block; *Figure 7*: exogenous NAD$^+$) would also lower spontaneous spiking in afferents and delay onset encoding.

## Ribeye and CtBP localization at synapses

In this study, we found that NAD(H) redox state had a dramatic effect on ribbon formation. NAD$^+$ promotes while NADH reduces ribbon size (*Figure 7*). The main component of ribbons is Ribeye. Ribeye has two domains, a unique A domain and a B domain that contains an NAD(H) binding domain (*Schmitz et al., 2000*). In vitro work on isolated A and B domains has shown that both

NAD$^+$ and NADH can affect interactions between A and B domains as well as interactions between B domains (*Magupalli et al., 2008*). In the context of ribbons, the B domain has been shown to concentrate at the interface between the ribbon and the membrane opposing the postsynapse (*Sheets et al., 2014*). Therefore, promoting B domain homodimerization may act to seed larger ribbons at the presynapse. In this scenario, NAD$^+$ and NADH could increase and decrease B domain homodimerization to impact ribbon size. We also observed an increase in cytoplasmic Ribeye aggregates after MCU block (*Figure 5F–G*). Therefore, it is alternatively possible that NAD$^+$ and NADH could impact interactions between A and B domains more broadly. NAD(H) redox could alter Ribeye interactions and alter the overall accumulation or separation of Ribeye within aggregates or at the presynapse.

Work in zebrafish has characterized lateral-line hair cells largely depleted of full-length Ribeye (*Lv et al., 2016*). When viewed using TEM, ribbons in Ribeye-depleted hair cells are strikingly transparent, suggesting that full-length Ribeye is required for the characteristic electron-dense structure of ribbons. Although these ribbons are smaller compared to controls, they are still able to tether vesicles near the active zone. Ribeye-depleted hair cells could be used to test whether mito-Ca$^{2+}$ and NAD(H) redox regulate ribbons size by impacting Ribeye interactions. If full-length Ribeye and its NAD(H) binding domain are the site of regulation, Ribeye-depleted hair cells would be unaffected by perturbations in mito-Ca$^{2+}$ and NAD(H).

Regardless of the exact mechanism, the effect of presynaptic activity, mito-Ca$^{2+}$ and related changes in NAD(H) redox homeostasis may extend beyond the sensory ribbon synapse. Ribeye is a splice variant of the transcriptional co-repressor CtBP2 (*Schmitz et al., 2000*). While the A domain is unique to Ribeye, the B domain is nearly identical to CtBP2 minus the nuclear localization sequence (NLS) (*Hübler et al., 2012*). In vertebrates, the CtBP family also includes CtBP1 (*Chinnadurai, 2007*). CtBP proteins are expressed in both hair cells and the nervous system, and there is evidence that both CtBP1 and CtBP2 may act as scaffolds at neuronal synapses. Interestingly, in cultured neurons, it has been shown that both synaptic activity and increased levels of NADH were associated with increased CtBP1 localization at the presynapse (*Ivanova et al., 2015*). In our in vivo study, we also found that the NAD$^+$/NADH ratio was lower (more NADH) in developing hair cells with intact presynaptic- and mito-Ca$^{2+}$ activities (*Figure 6H*). But in contrast to the in vitro work on CtBP1 in cultured neurons, we found that Ribeye localization to the presynapse and ribbon size were reduced when NADH levels were increased (*Figure 7A–C*). It is unclear why presynaptic activity regulates Ribeye localization differently from that of CtBP1. Ribeye and CtBP1 behavior may differ due to the divergent function of their N-terminal domains. Synaptic localization may also be influenced by external factors, such as the cell type in which the synapse operates, whether the study is performed in vitro or in vivo, as well as the maturity of the synapse. Overall, both studies demonstrate that the presynaptic localization of CtBP family members CtBP1 and Ribeye can be influenced by synaptic activity and NAD(H) redox state.

## Role of evoked mito-Ca$^{2+}$ uptake in mature hair cells

Sensory hair cells are metabolically demanding cells – both apical mechanotransduction and basal neurotransmission are energy demanding processes (*Shin et al., 2007*; *Spinelli et al., 2012*). Therefore, it is likely that hair-cell mitochondria play important roles in both of these functional domains. In mammalian auditory hair cells, mito-Ca$^{2+}$ uptake has been observed to buffer Ca$^{2+}$ beneath mechanosensory hair bundles (*Beurg et al., 2010*; *Fettiplace and Nam, 2019*). Blocking this uptake prolonged evoked Ca$^{2+}$ rises in hair bundles. This work suggested that apical mitochondria, along with the plasma membrane Ca$^{2+}$-ATPase (PMCA) contribute to cyto-Ca$^{2+}$ clearance to maintain optimal mechanotransduction (*Beurg et al., 2010*). Although the focus of our present study was on the synapse, we also found that blocking mito-Ca$^{2+}$ uptake using Ru360 (MCU antagonist) or TRO 19622 (VDAC antagonist) increased mechanosensitive-Ca$^{2+}$ responses in zebrafish lateral-line hair bundles (*Figure 2—figure supplement 1A–B'*). In the future it will be extremely interesting to explore the role apical mitochondria play in mechanotransduction.

In the presynaptic region of hair cells, the link between mito-Ca$^{2+}$ uptake and neurotransmission is less clear. Studies of synapses in various neuronal subtypes have demonstrated that mitochondria play multiple roles to maintain neurotransmission including: ATP production, Ca$^{2+}$ buffering and signaling, and neurotransmitter synthesis (reviewed in *Kann and Kovács, 2007*; *Vos et al., 2010*). A study on synaptic mitochondria at ribbon synapses in retinal-bipolar cells found that mito-Ca$^{2+}$

uptake was sporadic and did not significantly contribute to $Ca^{2+}$ clearance or the time course of evoked presynaptic-$Ca^{2+}$ responses (*Zenisek and Matthews, 2000*). This work concluded that mitochondria may contribute indirectly to $Ca^{2+}$ clearance from the synaptic terminal by providing ATP to fuel the PMCA. Our current work indicates that there is robust and reproducible mito-$Ca^{2+}$ uptake at the hair-cell presynapse during stimulation. But similar to work on retinal-bipolar cell ribbons, blocking mito-$Ca^{2+}$ uptake did not raise cyto-$Ca^{2+}$ levels, indicating it may not be critical for $Ca^{2+}$ clearance (*Figure 6A–C*). Instead, cyto-$Ca^{2+}$ levels may be maintained by the PMCA (*Beurg et al., 2010*; *Bortolozzi et al., 2010*). Alternatively, cyto-$Ca^{2+}$ levels may be maintained by the numerous $Ca^{2+}$ buffering proteins such as parvalbumin, calretinin, oncomodulin, calbindin and calmodulin that have been identified in hair cells (*Dechesne et al., 1991*; *Eybalin and Ripoll, 1990*; *Hackney, 2005*; *Pack and Slepecky, 1995*; *Pangršič et al., 2015*; *Rabié et al., 1983*; *Simmons et al., 2010*).

In our current study on basal, synaptic mitochondria, we found that in mature zebrafish-hair cells, mito-$Ca^{2+}$ uptake was critical for presynaptic-$Ca^{2+}$ influx. Even partial block of evoked mito-$Ca^{2+}$ uptake was sufficient to impair presynaptic-$Ca^{2+}$ influx, especially during sustained stimuli (*Figure 2C–D'*, *Figure 2—figure supplement 1D–E'*). Instead of buffering $Ca^{2+}$, our work indicates that mito-$Ca^{2+}$ uptake may impact $Ca_V1.3$-channel density (*Figure 2E–H*). In mature hair cells, after MCU block, impaired presynaptic-$Ca^{2+}$ responses coincided with an increase in $Ca_V1.3$-channel density at the presynapse (*Figure 2C–H*). Unfortunately, the majority of studies on $Ca_V1.3$ channels in hair cells focus on activity changes after a decrease or loss of $Ca_V1.3$-channel clustering. For example in Ribeye-depleted zebrafish hair cells $Ca_V1.3$ channels failed to cluster (*Lv et al., 2016*). In addition, when ribbons were enlarged in zebrafish-hair cells, $Ca_V1.3$-channel density was reduced (*Sheets et al., 2017*). In these studies, after a loss or reduction of $Ca_V1.3$-channel clustering, presynaptic-$Ca^{2+}$ signals were increased. Therefore, it is possible that an increase in $Ca_V1.3$-channel density could incur the opposite effect and decrease presynaptic-$Ca^{2+}$ responses. But how could an increase in $Ca_V1.3$-channel density decrease presynaptic-$Ca^{2+}$ responses? An increase in $Ca_V1.3$-channel density could enhance $Ca^{2+}$-dependent inactivation among tightly clustered $Ca_V1.3$ channels. In hair cells, $Ca_V1.3$ channels exhibit reduced $Ca^{2+}$ dependent inactivation (*Koschak et al., 2001*; *Platzer et al., 2000*; *Song et al., 2003*; *Xu and Lipscombe, 2001*). This reduction is thought to be important to transmit sustained sensory stimulation (*Kollmar et al., 1997*). Alternatively, an increase in $Ca_V1.3$-channel density could be a compensatory strategy to boost presynaptic activity after MCU block and impaired presynaptic-$Ca^{2+}$ influx. If channel density is not responsible for impaired presynaptic function, mito-$Ca^{2+}$ uptake could be critical to produce energy for other cellular tasks to maintain neurotransmission. Additional work is necessary to fully understand how evoked mito-$Ca^{2+}$ uptake functions to sustain presynaptic-$Ca^{2+}$ influx in mature zebrafish hair cells.

## Role of mito-$Ca^{2+}$ in metabolism versus pathology

In addition to a role in synapse function, mitochondria have been studied in the context of cellular metabolism and cell death (*Devine and Kittler, 2018*; *Tait and Green, 2013*; *Vakifahmetoglu-Norberg et al., 2017*). Our work suggests that mitochondria may play distinct roles in these processes in developing and mature hair cells. We found that mitochondria spontaneously take up $Ca^{2+}$ at the presynapse during hair-cell development (*Figure 4B–C*). Blocking presynaptic- and mito-$Ca^{2+}$ activities rapidly decreased the $NAD^+$/NADH ratio and altered ribbon size in developing hair cells (*Figures 5*, *6* and *7*). However, in mature hair cells, blocking these activities was pathological and did not influence NAD(H) redox (*Figure 6I*).

Some insight into these differences can be inferred from cardiac myocytes where the relationship between mito-$Ca^{2+}$ and NAD(H) redox has been extensively studied. Similar to our results in developing hair cells, in cardiac myocytes, mito-$Ca^{2+}$ drives cellular metabolism, which reduces $NAD^+$ to NADH (*Bertero and Maack, 2018*). In cardiac myocytes NADH is oxidized to $NAD^+$ when the MCU is blocked. These results are consistent with the changes in NAD(H) redox we observed in developing, but not mature hair cells. Instead, after complete MCU block in mature hair cells, we observed a loss of hair cells and synapses, and an increase in ribbon size (*Figure 3*). This outcome may be more similar to what occurs in heart failure or after extended MCU block – in cardiac myocytes, the production of oxidized $NAD^+$ quickly leads to energetic deficits, oxidative stress and ultimately the generation of reactive oxygen species (ROS) (*Bertero and Maack, 2018*). This is consistent with work in many cell types where mito-$Ca^{2+}$ loading is associated with pathological processes such as ROS production, cell death and synapse loss (*Cai and Tammineni, 2016*; *Court and Coleman, 2012*;

*DiMauro and Schon, 2008*; *Esterberg et al., 2013*; *Esterberg et al., 2014*; *Sheng and Cai, 2012*). Therefore, in mature hair cells, it is possible that after MCU block, changes in NAD(H) redox quickly become pathological. Recent work has suggested that younger hair cells may be more resilient to ototoxins, perhaps because they have not yet accumulated an excess of mitochondrial oxidation (*Pickett et al., 2018*). This could explain why complete MCU block alters NAD(H) redox without any observable pathological consequence in developing hair cells.

How could synapses be changing in mature hair cells during this pathology? It is possible that individual ribbons in mature hair cells are not enlarging, but instead ribbons are merging together as synapses are lost. Alternatively, remaining ribbons could be enlarging in order to compensate for loss of presynaptic function or synapses. In our current work, ribbon size was measured in fixed samples; therefore, it is difficult to distinguish between these possibilities. There is considerable work that suggests that synaptic structures, including ribbons are indeed dynamic structures (*Hull et al., 2006*; *Mehta et al., 2013*; *Sultemeier et al., 2017*). In the future, live imaging studies will help resolve whether there are different mechanisms underlying ribbon enlargement in mature and developing hair cells.

Overall the pathology observed in mature hair cells has parallels in recent work on noise-induced hearing that found measurable changes in ribbon morphology and synapse number following noise insult (*Jensen et al., 2015*; *Kujawa and Liberman, 2009*; *Liberman et al., 2015*). Work studying this type of hearing loss has shown that auditory inner hair cells in the high frequency region of the mouse cochlea have enlarged ribbons immediately after noise, followed later by synapse loss (*Liberman et al., 2015*). This pathology is reminiscent of our 1 hr pharmacological treatments that completely block the MCU in mature zebrafish-hair cells (*Figure 3E–F*). After this treatment, we observed a reduction in the number of hair cells and synapses, and an increase in ribbon size. Overall, these studies and our own data in mature hair cells support the association between mito-$Ca^{2+}$ and the MCU with pathological processes associated with ototoxins and noise-exposure.

In further support of this idea, recent work in mice has investigated the role of the MCU in noise-related hearing loss (*Wang et al., 2018*). This work demonstrated that pharmacological block or a loss of function mutation in MCU protected against synapse loss in auditory inner hair cells after noise exposure. Although this result is counter to our observed results where complete MCU block reduced synapse number (*Figure 3E*), it highlights an association between mito-$Ca^{2+}$, noise exposure and synapse integrity. It is possible that these differences can be explained by transitory versus chronic alterations in mito-$Ca^{2+}$ homeostasis. These differences may be resolved by studying hair cells in a zebrafish MCU knock out. In the future it will be interesting to examine both mito-$Ca^{2+}$ uptake and ribbon morphology during other pathological conditions that enlarge ribbons such as noise exposure, ototoxicity and aging.

Overall, our study has demonstrated the zebrafish-lateral line is a valuable system to study the interplay between the mitochondria, and synapse function, development and integrity. In the future it will be exciting to expand this research to explore how evoked and spontaneous mito-$Ca^{2+}$ influx are impacted by pathological treatments such as age, noise and ototoxins.

# Materials and methods

## Key resources table

| Reagent type (species) or resource | Designation | Source or reference | Identifiers | Additional information |
|---|---|---|---|---|
| Strain, strain background (*Danio rerio*) | Tübingen | ZIRC | RRID: ZIRC_ZL57; ZFIN ID: ZDB-GENO-990623–3 | |
| Strain, strain background (*Danio rerio*) | TL | ZIRC | RRID: ZIRC_ZL86; ZFIN ID: ZDB-GENO-990623–2 | |

*Continued on next page*

*Continued*

| Reagent type (species) or resource | Designation | Source or reference | Identifiers | Additional information |
|---|---|---|---|---|
| Genetic reagent (*Danio rerio*) | *Tg(myo6b: GCaMP6s-CAAX)<sup>idc1</sup>*; GCaMP6sCAAX; 6sCAAX; GCaMP6s | (*Jiang et al., 2017*) | RRID: ZFIN_ZDB-ALT-170113-3; https://zfin.org/ZDB-ALT-170113-3 | Membrane-localized calcium biosensor |
| Genetic reagent (*Danio rerio*) | *Tg(myo6b: RGECO1)<sup>yo10Tg</sup>*; CytoRGECO1 | (*Maeda et al., 2014*) | RRID: ZFIN_ZDB-ALT-150114-2; https://zfin.org/ZDB-ALT-150114-2 | Cell-fill calcium biosensor |
| Genetic reagent (*Danio rerio*) | *Tg(myo6b: GCaMP3)<sup>w78Tg</sup>*; CytoGCaMP3 | (*Esterberg et al., 2013*) | RRID: ZFIN_ZDB-ALT-130514-1; https://zfin.org/ZDB-ALT-130514-1 | Cell-fill calcium biosensor |
| Genetic reagent (*Danio rerio*) | *Tg(myo6b: mitoGCaMP3)<sup>w119Tg</sup>*; MitoGCaMP3; Mito | (*Esterberg et al., 2014*) | RRID: ZFIN_ZDB- ALT-141008–1; https://zfin.org/ZDB-ALT-141008-1 | Mitochondria-localized calcium biosensor |
| Genetic reagent (*Danio rerio*) | *Tg(myo6b:ribeye a-tagRFP)<sup>idc11Tg</sup>*; Rib a-RFP; Rib a-tagRFP; Rib a | (*Sheets, 2017*) | RRID: ZFIN_ZDB- ALT-190102–4; https://zfin.org/ZDB-ALT-190102-4 | Ribbon-localized fluorescent protein |
| Genetic reagent (*Danio rerio*) | *Tg(myo6b:mito RGECO1)<sup>idc12Tg</sup>*; MitoRGECO1 | This paper | RRID: ZFIN_ZDB-ALT-190102-5; https://zfin.org/ZDB-ALT-190102-5 | Mitochondria-localized calcium biosensor. See Materials and methods, 'Cloning and Transgenic Fish Production' |
| Genetic reagent (*Danio rerio*) | *Tg(myo6b: Rex-YFP)<sup>idc13Tg</sup>*; Rex-YFP | This paper | RRID: ZFIN_ZDB-ALT-190102-6; https://zfin.org/ZDB-ALT-190102-6 | Cell-fill NAD<sup>+</sup>/NADH ratio biosensor. See Materials and methods, 'Cloning and Transgenic Fish Production' |
| Antibody | Ribbon label: Mouse anti-Ribeye b IgG2a; Ribeye b; Ribeye; Rib b | (*Sheets et al., 2011*) | N/A | 1:10,000 |
| Antibody | PSD label: Mouse anti-pan-MAGUK IgG1; MAGUK | NeuroMab | RRID: AB_10673115; K28/86, #75–029 | 1:500 |
| Antibody | Hair cell label: Rabbit anti-Myosin VIIa | Proteus | RRID: AB_10015251; #25–6790 | 1:1000 |
| Antibody | goat anti-mouse IgG2a, Alexa Fluor 488 | ThermoFisher Scientific | RRID: AB_2535771; #A-21131 | 1:1000 |
| Antibody | goat anti-rabbit IgG (H+L) Alexa Fluor 568 | ThermoFisher Scientific | RRID: AB_143157; #A-11011 | 1:1000 |
| Antibody | goat anti-mouse IgG1 Alexa Fluor 647 | ThermoFisher Scientific | RRID: AB_2535809; #A-21240 | 1:1000 |
| Recombinant DNA reagent | Plasmid: 5E-pmyo6b | (*Kindt et al., 2012*) | N/A | |
| Recombinant DNA reagent | Plasmid: 3E-polyA | (*Kwan et al., 2007*) | #302 | |
| Recombinant DNA reagent | Plasmid: pDestTol2CG2 | (*Kwan et al., 2007*) | #395 | |
| Recombinant DNA reagent | Plasmid: pC1-Rex-YFP | (*Bilan et al., 2014*) | Addgene #48247 | |
| Recombinant DNA reagent | Plasmid: pDONR221 | ThermoFischer | Cat #12536017 | |

*Continued on next page*

*Continued*

| Reagent type (species) or resource | Designation | Source or reference | Identifiers | Additional information |
|---|---|---|---|---|
| Recombinant DNA reagent | Plasmid: CMV-R-GECO1 | (*Zhao et al., 2011*) | Addgene #32444 | |
| Recombinant DNA reagent | Plasmid: pME-Rex-YFP | This paper | N/A | See Materials and methods, 'Cloning and Transgenic Fish Production' |
| Recombinant DNA reagent | Plasmid: pME-mitoRGECO1 | This paper | N/A | See Materials and methods, 'Cloning and Transgenic Fish Production' |
| Sequence-based reagent | RexYFP attB FWD | This paper | PCR primers | GGGGACAAGTTTGTACAAAA AAGCAGGCTCCGCCACCATGA AGGTCCCCGAAGCG; Made by Integrated DNA Technologies (IDT). |
| Sequence-based reagent | REX-YFP attB REV | This paper | PCR primers | GGGGACCACTTTGTACAAGAA AGCTGGGTGTCACCCCAT CATCTCTTCCCG |
| Sequence-based reagent | RGECO1 FWD1 | This paper | PCR primers | [ATGTCCGTCCTGACGCCGCT GCTGCTGCGGGGCTTGACAGG CTCGGCCCGGCGGCTCCCAGT GCCGCGCGCCAA GATCCATTCGT TGGGGGATCCA]-GTCGACTCT TCACGTCGTAAGTG; Made by IDT. |
| Sequence-based reagent | RGECO1 REV1 | This paper | PCR primers | CTACTTCGCTGTCATCATTT GTACAAACTC; Made by IDT. |
| Sequence-based reagent | RGECO1 attB FWD2 | This paper | PCR primers | GGGGACAAGTTTGTACAAA AAAGCAGGCTGCCACCATGTC CGTCCTGACGCCGC; Made by IDT. |
| Sequence-based reagent | RGECO1 attB REV2 | This paper | PCR primers | GGGGACCACTTTGTACAAGA AAGCTGGGTGCTACTTCGC TGTCATCATTTGTACAAACTC; Made by IDT. |
| Peptide, recombinant protein | α-bungarotoxin | Tocris | 2133 | |
| Chemical compound, drug | Isradipine; Israd | Sigma-Aldrich | I6658 | |
| Chemical compound, drug | BayK 8644; BayK | Sigma-Aldrich | B133 | |
| Chemical compound, drug | Ru360 | Millipore | 557440 | |
| Chemical compound, drug | NAD⁺ | Sigma-Aldrich | N1511 | |
| Chemical compound, drug | NADH | Cayman Chemical | 16038 | |
| Chemical compound, drug | Tricaine; MESAB | Sigma-Aldrich | A5040 | |
| Chemical compound, drug | Paraformaldehyde: EM | Electron Microscopy Sciences | 15710 | |
| Chemical compound, drug | Glutaraldehyde: EM | Electron Microscopy Sciences | 16210 | |
| Chemical compound, drug | Paraformaldehyde: IHC | ThermoFisher Scientific | 28906 | |

*Continued on next page*

*Continued*

| Reagent type (species) or resource | Designation | Source or reference | Identifiers | Additional information |
|---|---|---|---|---|
| Chemical compound, drug | TRO 19622; TRO | Cayman Chemical | 21264 | Vortex vigorously to dissolve in embryo media + 0.1% DMSO |
| Software, algorithm | Prism (v. 8) | Graphpad Software | RRID: SCR_002798; https://www.graphpad.com | |
| Software, algorithm | Adobe Illustrator | Adobe | RRID: SCR_014198; https://www.adobe.com | |
| Software, algorithm | FIJI is just ImageJ | NIH | RRID: SCR_003070; https://fiji.sc | |
| Software, algorithm | Zen | Zeiss | RRID: SCR_01367; https://www.zeiss.com/microscopy/int/products/microscope-software/zen.html | |
| Software, algorithm | Prairie View | Bruker Corporation | RRID: SCR_017142; https://www.bruker.com/products/fluorescence-microscopes/ultima-multiphoton-microscopy/ultima-in-vitro/overview.html | |
| Software, algorithm | Python Programming Language | Python Software Foundation | RRID: SCR_008394; https://www.python.org/ | |

## Zebrafish husbandry and genetics

Adult *Danio rerio* (zebrafish) were maintained under standard conditions. Larvae 2 to 6 days post-fertilization (dpf) were maintained in E3 embryo medium (in mM: 5 NaCl, 0.17 KCl, 0.33 $CaCl_2$ and 0.33 $MgSO_4$, buffered in HEPES pH 7.2) at 28°C. All husbandry and experiments were approved by the NIH Animal Care and Use program under protocol #1362–13. Transgenic zebrafish lines used in this study include: *Tg(myo6b:GCaMP6s-CAAX)*$^{idc1}$ (*Jiang et al., 2017*), *Tg(myo6b:RGECO1)*$^{vo10Tg}$ (*Maeda et al., 2014*), *Tg(myo6b:GCaMP3)*$^{w78Tg}$ (*Esterberg et al., 2013*), *Tg(myo6b:mitoGCaMP3)*$^{w119Tg}$ (*Esterberg et al., 2014*), and *Tg(myo6b:ribeye a-tagRFP)*$^{idc11Tg}$ (*Sheets, 2017*). Experiments were performed using Tübingen or TL wildtype strains.

## Cloning and transgenic fish production

To create transgenic fish, plasmid construction was based on the tol2/Gateway zebrafish kit developed by the lab of Chi-Bin Chien at the University of Utah (*Kwan et al., 2007*). These methods were used to create *Tg(myo6b:mitoRGECO1)*$^{idc12Tg}$ and *Tg(myo6b:Rex-YFP)*$^{idc13Tg}$ transgenic lines. Gateway cloning was used to clone *Rex-YFP* (*Bilan et al., 2014*) and *mitoRGECO1* into the middle entry vector pDONR221. For mitochondrial matrix targeting, the sequence of cytochrome C oxidase subunit VIII (*Rizzuto et al., 1989*) was added to the N-terminus of RGECO1. Vectors p3E-polyA (*Kwan et al., 2007*) and pDestTol2CG2 (*Kwan et al., 2007*) were recombined with p5E-*myosinVIb* (*myo6*) (*Kindt et al., 2012*) and our engineered plasmids to create the following constructs: *myo6b:REX-YFP* and *myo6b:mitoRGECO1*. To generate transgenic fish, DNA clones (25–50 ng/μl) were injected along with *tol2* transposase mRNA (25–50 ng/μl) into zebrafish embryos at the single-cell stage.

## Pharmacological treatment of larvae for immunohistochemistry

For pharmacological studies, zebrafish larvae were exposed to compounds diluted in E3 with 0.1% DMSO (Isradipine, Bay K8644, NAD$^+$ (Sigma-Aldrich, St. Louis, MO), Ru360 (Millipore, Burlington, MA), TRO 19622 (Cayman Chemical, Ann Arbor, MI)) or Tris-HCl (NADH (Cayman Chemical, Ann Arbor, MI)) for 30 min or 1 hr at the concentrations indicated. E3 with 0.1% DMSO or Tris-HCl were used as control solutions. In solution at pH 7.0–7.3, NADH oxidizes into NAD$^+$ by exposure to dissolved oxygen. To mitigate this, NADH was dissolved immediately before use and was exchanged

with a freshly dissolved NADH solution every half hour. Dosages of isradipine, Ru360, Bay K8644, TRO 19622, $NAD^+$ and NADH did not confer excessive hair-cell death or synapse loss unless stated. After exposure to the compounds, larvae were quickly sedated on ice and transferred to fixative.

## In vivo imaging of baseline $Ca^{2+}$ and NAD(H) redox

To prepare larvae for imaging, larvae were immobilized as previously described (*Kindt et al., 2012*). Briefly, larvae were anesthetized with tricaine (0.03%) in E3 and pinned to a chamber lined with Sylgard 184 Silicone Elastomer (Dow Corning, Midland, MI). Larvae were injected with 125 μM α-bungarotoxin (Tocris, Bristol, UK) into the pericardial cavity to induce paralysis. Tricaine was rinsed off the larvae and replaced with fresh E3.

For baseline measurements of Rex-YFP and CytoRGECO1 fluorescence, larvae were imaged using an upright Nikon ECLIPSE Ni-E motorized microscope (Nikon Inc, Tokyo, Japan) in widefield mode with a Nikon 60 × 1.0 NA water-immersion objective, an 480/30 nm excitation and 535/40 nm emission filter set or 520/35 nm excitation and 593/40 emission filter set, and an ORCA-D2 camera (Hamamatsu Photonics K.K., Hamamatsu City, Japan). Acquisitions were taken at 5 Hz, in 15 plane Z-stacks every 2 μm. For baseline measurements of MitoGCaMP3, larvae were imaged using a Bruker Swept-field confocal microscope (Bruker Inc, Billerica, MA), with a Nikon CFI Fluor 60 × 1.0 NA water-immersion objective. A Rolera EM-C2 CCD camera (QImaging, Surrey, Canada) was used to detect signals. Acquisitions were taken using a 70 μm slit at a frame rate of 10 Hz, in 26 plane Z-stacks every 1 μm. MitoGCaMP3 baseline intensity varied dramatically in controls between timepoints. To offset this variability, we acquired and averaged the intensity of 4 Z-stacks per time point. For all baseline measurements transgenic larvae were first imaged in E3 with 0.1% DMSO or 0.1% Tris-HCl as appropriate. Then larvae were exposed to pharmacological agents for 30 min and a second acquisition was taken. Any neuromasts with cell death after pharmacological or mock treatment were excluded from our analyses.

## In vivo imaging of evoked $Ca^{2+}$ signals

To measure evoked $Ca^{2+}$ signals in hair cells, larvae were immobilized in a similar manner as described for baseline measurements. After α-bungarotoxin paralysis, larvae were immersed in neuronal buffer solution (in mM: 140 NaCl, 2 KCl, 2 $CaCl_2$, 1 $MgCl_2$ and 10 HEPES, pH 7.3). Evoked $Ca^{2+}$ measurements were acquired using the Bruker Swept-field confocal system described above. To stimulate lateral-line hair cells, a fluid-jet was used as previously described to deliver a saturating stimulus (*Lukasz and Kindt, 2018*).

To measure presynaptic GCaMP6sCAAX signals at ribbons, images were acquired with 1 × 1 binning using a 35 μm slit at 50 Hz in a single plane containing presynaptic ribbons (*Figure 2—figure supplement 1C–C'*). Ribbons were marked in live hair cells using the *Tg(myo6b:ribeye a-tagRFP)*[idc11Tg] transgenic line (*Figure 2—figure supplement 1C*). Ribbons were located relative to GCaMP6s signals by acquiring a 2-color Z-stack of 5 planes every 1 μm at the base of the hair cells. To correlate presynaptic GCaMP6sCAAX signals with MitoRGECO1 signals in hair cells, 2-color imaging was performed. Images were acquired in a single plane with 2 × 2 binning at 10 Hz with a 70 μM slit. MitoGCaMP3 signals were acquired at 10 Hz in Z-stacks of 5 planes 1 μm apart with 2 × 2 binning and a 70 μM slit. High speed imaging along the Z-axis was accomplished by using a piezoelectric motor (PICMA P-882.11–888.11 series, Physik Instrumente GmbH, Karlsruhe, Germany) attached to the objective to allow rapid imaging at a 50 Hz frame rate yielding a 10 Hz volume rate. Due to the slow mito-$Ca^{2+}$ return to baseline after stimulation (~5 min), we waited a minimum of 5 min before initiating a new evoked GCaMP6sCAAX or MitoGCaMP3 acquisition. To examine mechanotransduction, GCaMP6sCAAX signals were measured in apical hair bundles (*Figure 2—figure supplement 1A–B'*; *Zhang et al., 2018*). Apical GCaMP6sCAAX signals were acquired in a single plane at 1 × 1 binning with a 35 μM slit at 20 Hz. For pharmacological treatment, acquisitions were made prior to drug treatment and after a 30 min incubation in the pharmacological agent. Any neuromasts with cell death after pharmacological treatment were excluded from our analyses.

## In vivo imaging of spontaneous $Ca^{2+}$ signals

To measure spontaneous $Ca^{2+}$ signals in hair cells, larvae were prepared in a similar manner as described for evoked $Ca^{2+}$ measurements. Spontaneous $Ca^{2+}$ measurements were acquired using

the Bruker Swept-field confocal system described above. To measure spontaneous presynaptic GCaMP6sCAAX signals, images were acquired with 2 × 2 binning with a 70 µm slit at 0.33 Hz in a single plane for 900 s. For acquisition of two-color spontaneous presynaptic GCaMP6sCAAX and MitoRGECO1 signals images were acquired with 2 × 2 binning with a 70 µm slit at 0.2 Hz in a single plane for 900 s.

## Electron microscopy

Larvae were prepared for electron microscopy as described previously (*Sheets et al., 2017*). Transverse serial sections (~60 nm thin sections) were used to section through neuromasts. Samples were imaged on a JEOL JEM-2100 electron microscope (JEOL Inc, Tokyo, Japan). The distance from the edge of a ribbon density to the edge of the nearest mitochondrion was measured (n = 17 ribbons). A subset of measurements was taken from more than one ribbon within a hair cell. At 81% of ribbons, a mitochondrion could be clearly identified within 1 µm of a ribbon (17 out of 21 ribbons). All distances and perimeters were measured in FIJI (*Schindelin et al., 2012*).

## Immunofluorescence staining and airyscan imaging

Whole larvae were fixed with 4% paraformaldehyde in PBS at 4°C for 3.5–4 hr as previously described (*Zhang et al., 2018*). Fixative was washed out with 0.01% Tween in PBS (PBST) in four washes, 5 min each. Larvae were then washed for 5 min with $H_2O$. The $H_2O$ was thoroughly removed and replaced with ice-cold acetone and placed at $-20°C$ for 3 min for 3 dpf and 5 min for 5 dpf larvae, followed by a 5 min $H_2O$ wash. The larvae were then washed for 4 × 5 min in PBST, then incubated in block overnight at 4°C in blocking solution (2% goat serum, 1% bovine serum albumin, 2% fish skin gelatin in PBST). Primary and secondary antibodies were diluted in blocking solution. Primary antibodies and their respective dilutions are: Ribbon label: Mouse anti-Ribeye b IgG2a, 1:10,000 (*Sheets et al., 2011*); PSD label: Mouse anti-pan-MAGUK IgG1, 1:500 (MABN72, Millipore-Sigma, Burlington, MA); Hair-cell label: Rabbit anti-Myosin VIIa, 1:1000 (#25–6790, Proteus BioSciences Inc, Ramona, CA); $Ca_V1.3$ channel label: Rabbit anti-$Ca_V$1.3a, 1:500 (*Sheets et al., 2012*). Larvae were incubated in primary antibody solution for 2 hr at room temperature. After 4 × 5 min washes in PBST to remove the primary antibodies, diluted secondary antibodies were added and samples were incubated for 2 hr at room temperature. Secondary antibodies and their respective dilution are as follows: goat anti-mouse IgG2a, Alexa Fluor 488, 1:1000; goat anti-rabbit IgG (H+L) Alexa Fluor 546, 1:1000; goat anti-mouse IgG1 Alexa Fluor 647, 1:1000 (Thermo Fisher Scientific, Waltham, MA). Secondary antibodies were washed out with PBST for 3 × 5 min, followed by a 5 min wash with $H_2O$. Larvae were mounted on glass slides with Prolong Gold Antifade Reagent (Invitrogen, Carlsbad, CA) using No. 1.5 coverslips.

Prior to Airyscan imaging, live samples were immobilized in 2% low-melt agarose in tricaine (0.03%) in cover-glass bottomed dishes. Live and fixed samples were imaged on an inverted Zeiss LSM 780 laser-scanning confocal microscope with an Airyscan attachment (Carl Zeiss AG, Oberkochen, Germany) using an 63 × 1.4 NA oil objective lens. The median (±median absolute deviation) lateral and axial resolution of the system was measured at 198 ± 7.5 nm and 913 ± 50 nm (full-width at half-maximum), respectively. The acquisition parameters were adjusted using the control sample such that pixels for each channel reach at least 1/10 of the dynamic range. The Airyscan Z-stacks were processed with Zeiss Zen Black software v2.1 using 3D filter setting of 7.0. Experiments were imaged with the same acquisition settings to maintain consistency between comparisons.

## Analysis of Ca²⁺ and NAD(H) signals, processing, and quantification

To quantify changes in baseline $Ca^{2+}$ and NAD(H) homeostasis, images were processed in FIJI. For our measurements we quantified the fluorescence in the basal-most 8 µm (four planes) to avoid overlap between cells. The basal planes were max Z-projected, and a 24.0 µm (Rex-YFP and RGECO1) or 26.8 µm (MitoGCaMP3) circular region of interest (ROI) was drawn over the neuromast to make intensity measurements. To correct for photobleaching, a set of mock-treated control neuromasts were imaged during every trial. These mock treatments were used to normalize the post-treatment intensity values.

To quantify the magnitude of evoked changes in $Ca^{2+}$, images were processed in FIJI. Images in each time series were aligned using Stackreg (*Thévenaz et al., 1998*). For evoked MitoRGECO1,

MitoGCaMP3, CytoGCaMP3 and two-color GCaMP6sCAAX and MitoRGECO1 signals, Z-stacks were max z-projected, and a 5 μm diameter circular ROI was drawn over each hair cell to make intensity measurements. For ribbon-localized measurements, GCaMP6sCAAX signals were measured within 1.34 μm round ROIs at individual ribbons, and intensity change at multiple ribbons per cell were averaged. For measurements of mechanotransduction, GCaMP6sCAAX signals were measured within 1.34 μm round ROIs at individual hair bundles, and intensity change in multiple bundles per neuromast were averaged.

To plot evoked changes in $Ca^{2+}$, we subtracted the baseline ($F_0$, signal during the pre-stimulus period) was subtracted from each timepoint acquired. Then each timepoint was divided by $F_0$ to generate the relative change in fluorescent signal from baseline or $\Delta F/F_0$. Quantification of evoked $Ca^{2+}$ signals were made on max $\Delta F/F_0$ measurements. Cells with presynaptic-$Ca^{2+}$ activity are defined by max $\Delta F/F_0$ of >0.05 for MitoRGECO1 and MitoGCaMP3, and max $\Delta F/F_0$ >0.25 for GCaMP6sCAAX for a 2 s stimulation. The method to obtain and overlay the spatial signal distribution of evoked signals as heat maps has been previously described (*Lukasz and Kindt, 2018*). We first computed the baseline image ($F_0$ or reference image) by averaging the images over the pre-stimulus period. Then the baseline image ($F_0$) was subtracted from each image acquired, to represent the relative change in fluorescent signal from baseline or $\Delta F$. The $\Delta F$ signal images during the stimulus period were binned, scaled and encoded by color maps with red indicating an increase in signal intensity.

To quantify the average magnitude and frequency of spontaneous $Ca^{2+}$ changes in GCaMP6sCAAX signals, images were processed in Matlab R2014b (Mathworks, Natick, MA) and ImageJ (NIH, Bethesda, MD). First, images in each time series were aligned in ImageJ using Stackreg (*Thévenaz et al., 1998*). To measure the average magnitude during the 900 s GCaMP6sCAAX image acquisition, a 5 μm diameter circular ROI was drawn over each hair cell and a raw intensity value was obtained from each time point. Then, in Matlab, the raw traces were bleach corrected. Next, the corrected intensity values were normalized as $\Delta F/F_0$. For spontaneous $Ca^{2+}$ signals $F_0$ is defined as the bottom 15th percentile of fluorescence values (*Babola et al., 2018*). Then, values of $\Delta F/F_0$ of less than 10% were removed. These values were considered to be noise and our threshold value for a true signal. A 10% threshold was determined by imaging spontaneous GCaMP6CAAX signals in the presence of isradipine where no signals were observed (*Figure 4—figure supplement 1*). The averaged magnitude of spontaneous activity per cell was obtained by dividing the integral/sum of GCaMP6sCAAX signals ($\Delta F/F_0$ >10%) during the whole recording period by 300 (300 frames in 900 s). The frequency of GCaMP6sCAAX signals was defined as the average number of peaks per second during the whole recording period.

## Image processing and quantification of synapse morphology

To quantify synapse morphology and pairing, images were first processed in ImageJ, and then synapses were paired using Python (Python Software Foundation, Wilmington, DE) in the Spyder Scientific Environment (MIT, Cambridge, MA). In ImageJ, each Airyscan Z-stack was background subtracted using rolling-ball subtraction. Z-stacks containing the MAGUK channel were further band-pass filtered to remove details smaller than six px and larger than 20 px. A duplicate of each Z-stack was normalized for intensity. This duplicated Z-stack was used to identify individual ribbon and MAGUK using the Simple 3D Segmentation of ImageJ 3D Suite (*Ollion et al., 2013*). Local intensity maxima, identified with 3D Fast Filter, and 3D watershed were used to separate close-by structures. The centroids for each identified ribbon and MAGUK puncta were obtained using 3D Manager and these coordinates were used to identify complete synapses. The max Z-projection of the segmented Z-stack was used to generate a list of 2D objects as individual ROIs corresponding to each punctum. This step also included a minimum size filter: Ribeye: 0.08 μm², MAGUK: 0.04 μm². For quantification of extrasynaptic Ribeye b puncta, the minimum size filter was not applied. The 2D puncta ROI were applied over the max Z-projection of the original Z-stack processed only with background subtraction. This step measures the intensity of the antibody label. Centroid and intensity information were exported as a CSV spreadsheet (macro is available on https://github.com/wonghc/ImageJ-ribbon-synapse-quantification, *Wong, 2019*; copy archived at https://github.com/elifesciences-publications/ImageJ-ribbon-synapse-quantification).

In Python, the 3D centroid coordinates for each ribbon punctum were measured against the coordinates of every post-synaptic MAGUK punctum to find the MAGUK punctum within a threshold

distance. This threshold was calculated by taking the 2D area of the Ribeye and MAGUK punctum measured in the max Z-projection to calculate an approximate radius by dividing by π and taking the square root. The two radii were then summed to get the threshold. Puncta that were not paired were excluded from later statistical analyses of synaptic ribbon and postsynaptic MAGUK puncta. To quantify the amount of $Ca_V1.3$ immunolabel at ribbons, 2D ROIs generated from the Ribeye label to generate ribbon areas were applied to a max Z-projection of the $Ca_V1.3$ immunolabel. The integrated intensity of $Ca_V1.3$ immunolabel was measured within each ROI. The number of hair cells, synapses per cell, and $Ca_V1.3$ clusters per PSD were counted manually. Hair-cell counts were assayed with Myosin VIIa antibody label in treatments when synapse or cell numbers were reduced. Due to slight variability between clutches and immunostains we only compared experimental data taken from the same clutch, immunostain and imaging session.

## Statistics

Statistical analyses and data plots were performed with Prism 8 (Graphpad, San Diego, CA). Values of data with error bars on graphs and in text are expressed as mean ± SEM unless indicated otherwise. All experiments were performed on a minimum of 2 animals, 6 neuromasts (posterior lateral-line neuromasts L1-L4 or anterior lateral-line neuromasts O1 and O2 [*Figure 1—figure supplement 2*, *Figure 3—figure supplement 1*, *Figure 5—figure supplement 1*]), on two independent days. For 3 and 5 dpf larvae each neuromast represents analysis from 8 to 12 hair cells; 24–36 synapses and 14–18 hair cells; 42–54 synapses respectively. All replicates are biological. Based on the variance and effect sizes reported previously and measured in this study, these numbers were adequate to provide statistical power to avoid both Type I and Type II error (*Sheets et al., 2012*; *Zhang et al., 2018*). No animals or samples were excluded from our analyses unless control experiments failed–in these cases all samples were excluded. No randomization or blinding was used for our animal studies. Where appropriate, data was confirmed for normality using a D'Agostino-Pearson normality test and for equal variances using a F test to compare variances. Statistical significance between two conditions was determined by either an unpaired *t*-test, an unpaired Welch's unequal variance *t*-test, a Mann-Whitney U test or a Wilcoxon matched-pairs signed-rank test as appropriate. For comparison of multiple conditions, a Brown-Forsythe with Dunnett's T3 post hoc or a Brown-Forsythe and Welch ANOVA with Holm-Sidak's post hoc were used as appropriate. To calculate the $IC_{50}$ for Ru360 block of evoked MitoGCaMP3 signals a dose response curve was plotted using 0, 0.5, 2, 5 and 10 μM Ru360. A non-linear fit with four parameters and a variable slope was performed to calculate an $IC_{50}$ of 1.37 μM.

## Acknowledgements

This work was supported by National Institute on Deafness and Other Communication Disorders Intramural Research Program Grant 1ZIADC000085-01 to KSK and ZICDC000081 to RSP and Y-XW. We would like to thank Daria Lukasz, Katie Drerup, Paul Fuchs and Doris Wu for their support and thoughtful comments on the manuscript.

## Additional information

### Funding

| Funder | Grant reference number | Author |
| --- | --- | --- |
| National Institute on Deafness and Other Communication Disorders | 1ZIADC000085-01 | Hiu-tung C Wong<br>Qiuxiang Zhang<br>Alisha J Beirl<br>Katie Kindt |
| National Institute on Deafness and Other Communication Disorders | ZICDC000081 | Ronald S Petralia<br>Ya-Xian Wang |

The funders had no role in study design, data collection and interpretation, or the decision to submit the work for publication.

## Author contributions

Hiu-tung C Wong, Conceptualization, Data curation, Software, Formal analysis, Investigation, Methodology, Writing—original draft, Writing—review and editing; Qiuxiang Zhang, Alisha J Beirl, Ronald S Petralia, Formal analysis, Investigation; Ya-Xian Wang, Data curation, Formal analysis; Katie Kindt, Conceptualization, Data curation, Formal analysis, Supervision, Funding acquisition, Investigation, Methodology, Writing—original draft, Project administration, Writing—review and editing

## Author ORCIDs

Hiu-tung C Wong  https://orcid.org/0000-0001-5826-8526
Katie Kindt  https://orcid.org/0000-0002-1065-8215

## Ethics

Animal experimentation: All husbandry and experiments were approved by the NIH Animal Care and Use program under protocol #1362-13.

## Decision letter and Author response

Decision letter https://doi.org/10.7554/eLife.48914.sa1
Author response https://doi.org/10.7554/eLife.48914.sa2

# Additional files

## Supplementary files

• Transparent reporting form

## Data availability

Source data has been provided for all figures and figure supplements.

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
