## [Decision Letter]

Thank you for submitting your article "Synaptic mitochondria are critical for hair-cell synapse formation and function" for consideration by *eLife*. Your article has been reviewed by three peer reviewers, and the evaluation has been overseen by a Reviewing Editor and Andrew King as the Senior Editor. The following individuals involved in review of your submission have agreed to reveal their identity: Hernán López-Schier (Reviewer #1); Anna Lysakowski (Reviewer #3).

The reviewers have discussed the reviews with one another and the Reviewing Editor has drafted this decision to help you prepare a revised submission.

Summary:

Wong and colleagues present an interesting study of the effects of mitochondrial calcium handling on ribbon synapse structure in zebrafish lateral line hair cells. Using a combination of techniques, the authors provide evidence that mechanotransduction-driven calcium influx results in increased mitochondrial calcium uptake, in turn altering ribbon size. The authors also provide evidence for ribbon regulation by changes in NAD^+^/NADH driven by mitochondrial calcium changes. The work is clearly presented and has important implications for regulation of synaptic function in normal and damaged hair cells.

Essential revisions:

1) The reviewers agreed that use of an alternative inhibitor of mitochondrial calcium uptake will strengthen the results (see comments from reviewer 2).

2) There was some confusion over the inclusion of immature cells in Figure 2 (see comments from reviewer 2). Please clarify.

3) Tone down conclusions or interpretation where requested (see reviews for details), or provide further clarification or explanation.

The full reviews are appended below for your information.

Reviewer #1:

In this manuscript, Wong and colleagues analyse the role of mitochondria in hair-cell synaptogenesis in zebrafish. They show that presynaptic Ca^2+^ influx drives Ca^2+^ uptake by mitochondria. In mature hair cells, mito-Ca^2+^ uptake occurs during evoked stimulation and is required to sustain presynaptic function. In developing hair cells, decreasing mito-Ca^2+^ levels increases the NAD^+^/NADH redox ratio, which modify ribbon formation.

Subsection “Mitochondria are located near presynaptic ribbons”, last paragraph: The authors find a mitochondrion in proximity to ribbons in 17 out of 21 sections. Are some of these sections of the same hair cells, or do they come from 21 independent hair cells?

Next they claim that in lateral-line hair cells, mitochondria are present near ribbons. However, the images shown clearly reveal that mitochondria are everywhere in hair cells, which unsurprisingly will include the areas near ribbons. Therefore, the claim that mitochondria are poised to impact ribbon synapses is overreaching.

The data shown cannot be used to claim that "MitoGCaMP3 signals propagated apically within the mitochondria". Instead, MitoGCaMP3 signals propagated apically. The use of Ru360 does not directly support the claim either, because the authors would be blocking Ca^2+^ entry into mitochondria generally from any source including cytoplasm or the ER, and not a spread of within the mitochondria.

I am surprised with what appears to be zero MitoGCaMP3 signal outside the active-zone area at the start of the 2-s mechanical stimulation of neuromasts, given the long perdurance of the signal after stimulation (5 min to return to baseline). Hair cells must have been stimulated constantly during the experiment. Otherwise, the authors may want to explain this observation and how the prevented hair-cell activity before the 2-s evoked stimulation.

Figure 3F: Are Ru360 2 μM for 1hr and DMSO for Ru360 10 μM for 1hr significantly different?

The authors measure and compare Ribeye b area assuming that the 3D shape of the ribbons is identical under every treatment. Have they confirmed the case?

I am not sure what the authors refer to with "spontaneous Ca_V_1.3 and MCU Ca^2+^ activities".

I am surprised that the authors do not discuss their findings in relation to the findings of Zenisek and Matthews, 2000.

In subsection “Functional significance of ribbon size”, the authors state: "Functionally, compared to smaller ribbons, larger ribbons are associated with afferent fibers with less spontaneous activity and higher thresholds of activation". What fibers are the authors referring to? Afferent fibers of the lateral line do not show intrinsic spontaneous activity. Their spontaneous activity is given by spontaneous release of glutamate from the hair cells.

Is there any indication that altered Ca^2+^ entry into mitochondria alters the localization of Ca_V_1.3 channels?

The authors must discuss their findings in relation to Lv et al., 2016.

Reviewer #2:

Beurg et al., 2010, provide evidence for mitochondrial responses to apical calcium entry in cultured mammalian hair cells. These results contrast to the findings of the current work, where calcium spreads from basal mitochondria and are blocked by the calcium channel blocker isradipine. Is this due to differences in the kinetics of stimulation or differences in the type of hair cells?

Results displayed in Figure 2 are described as in mature hair cells but it appears some of the data were from 3 dpf. It is not clear why immature cells are included given the later results showing age-dependent differences.

It is somewhat surprising that blocking calcium uptake into mitochondria reduces presynaptic calcium signals. It is potentially possible that Ru360 may have non-specific effects such as blocking mechanotransduction. This could be tested by examining apical calcium signals, which the Kindt lab has found occur in all hair cells in response to mechanotransduction. An alternative inhibitor of mitochondrial calcium uptake such as VDAC inhibitors might also strengthen this result.

Throughout the authors use the term "formation" to describe effects on synapses. It might be slightly pedantic, but it appears that the effects they observe do not affect formation (no changes in the number of synapses; Figure 5D, 7D) but rather size and shape (Figure 5E, 7E). Formation implies the localization of new synapses rather than alteration of synapses already formed by other mechanisms.

It is puzzling that blocking mitochondrial calcium accumulation results in an increase in NAD^+^. Mitochondrial oxidation of NADH to promote NAD^+^ production is usually thought to be stimulated by calcium, which is the opposite effect of the one observed here. Given the puzzling nature of the link between calcium and NAD^+^, could additional experiments be performed to strengthen the results? Could uncoupling of mitochondria have an effect on ribbon size? Could pyruvate accumulation be the result? Could similar effects be seen with the MPC inhibitor UK5099?

Reviewer #3:

This is an extremely interesting and well-written paper, with excellent, beautiful data. The authors make a compelling case for the effects of mitochondrial Ca^2+^ regulating synapse size and number in hair cells during development. I congratulate them on this excellent study. That said, I do have several comments that I hope will improve this beautiful paper.

Subsection “Mito-Ca^2+^ uptake at ribbons is MCU and Ca_V_1.3 dependent”, last paragraph: The authors claim that the effect of Ru360 is dose-dependent. Were other doses used? If so, what was the cut-off dose for the effect, i.e., what was the IC_50_ or EC_50_?

Subsection “Mito-Ca^2+^ uptake occurs in cells with presynaptic Ca^2+^ influx”, first paragraph: It has been described that approximately 10-40% of PSDs in mammalian hair cells do not have ribbons and about a third of boutons do not have ribbons (Sadeghi et al., 2014; Lysakowski and Goldberg, 1997). This is roughly equal to the 40% of cells in the present paper in which mito-Ca^2+^ uptake was present and the 30% of cells in the Zhang et al., 2018, paper that are synaptically active. I wonder if this is a "snapshot in time" of ribbon turnover, which has been documented in pinealocytes, but has not so far been described in hair cells. There were, in fact, a couple of papers on circadian rhythms affecting ribbon numbers in cochlear hair cells (Swetlitschkin and Vollrath, 1988; Molina Mira and Martinez-Soriano, 1991) cited in this review (Wagner, Cell Tissue Res. 1997, PMID: 9027298), which will likely be interesting to the authors, as the review also discusses developmental changes in ribbon synapse production due to Ca^2+^ effects.

Subsection “Evoked mito-Ca^2+^ uptake is important for mature synapse integrity and cell health”, last paragraph: This study was done in posterior neuromasts. I wonder if it is known whether there are differences between anterior and posterior neuromasts in terms of sensitivity or ribbon size? I also wonder if the ribbons are becoming bigger to compensate for the loss of synapses. In the microgravity of spaceflight, ribbons can become more numerous, perhaps compensating for the loss of gravity (Ross, Acta Otolaryngol, 2000) or less numerous (Sultemeier et al., 2017).

Subsections “Spontaneous mito-Ca^2+^ uptake regulates ribbon formation”, second paragraph and “Role of spontaneous mito-Ca^2+^ uptake in developing hair cells”: The authors state that in developing hair cells, after application of Ru360, there is an increase in ribbon size. There is also, apparently, an increase in the number of cytoplasmic Ribeye-positive inclusions. How do we know that the size increase is not just that the ribbon clusters are not separating? Normally, ribbons are produced and "float" within the cytoplasm in clusters, presumably being transported through the cytoplasm by microtubules (this is supported by the observable Ribeye-positive cytoplasmic inclusions), then they separate into individual ribbons near and at the plasma membrane. Again, this may only be discriminable by TEM.

Subsection “MCU and Ca_V_1.3 channel activities regulate subcellular Ca^2+^ homeostasis”, last paragraph: Another possibility that doesn't seem to be considered in the paper or in the Discussion is that there are cytoplasmic calcium binding proteins that also buffer the Ca^2+^.

Subsection “Functional significance of ribbon size”, second paragraph: The synaptic ribbons in central/striolar type II hair cells feeding into irregular, dimorphic vestibular afferents are larger than those feeding into peripheral regular, dimorphic afferents (Lysakowski and Goldberg, 1997). Similarly, irregular afferents have lower spontaneous background activity than regular afferents (Table 2, Goldberg, Smith and Fernández, 1984). So, there is some similarity to cochlear afferents.

---

## [Author Response]

Essential revisions:1) The reviewers agreed that use of an alternative inhibitor of mitochondrial calcium uptake will strengthen the results (see comments from reviewer 2).

We have used a second inhibitor of mitochondrial calcium uptake to support our pharmacological results in mature hair cells. Similar to the MCU inhibitor Ru360, the VDAC inhibitor TRO 19622 (1) blocks mitochondrial calcium uptake and (2) impairs presynaptic calcium signals.

2) There was some confusion over the inclusion of immature cells in Figure 2 (see comments from reviewer 2). Please clarify.

Regarding Figure 2A-B: Although there are synaptically active mature hair cells at 3 dpf, we agree using immature cells in this figure is confusing as the focus is on mature hair cells. Therefore, Figure 2 has been modified to include an example of mature hair cells (5 dpf).

3) Tone down conclusions or interpretation where requested (see reviews for details), or provide further clarification or explanation.

Our conclusion section has been modified to accommodate the reviewers’ suggestions, and we have toned down some of our own interpretations and conclusions as requested.

The full reviews are appended below for your information.Reviewer #1:[…]Subsection “Mitochondria are located near presynaptic ribbons”, last paragraph: The authors find a mitochondrion in proximity to ribbons in 17 out of 21 sections. Are some of these sections of the same hair cells, or do they come from 21 independent hair cells?

Two of the measurements were taken from multiple ribbons in a single hair cell. We have changed our N to reflect ribbons rather than sections in the Results, Materials and methods and Figure 1 legend.

Results – “Using TEM, we examined sections that clearly captured ribbons (example, Figure 1C). Near the majority of ribbons (81% ) we observed a mitochondrion in close proximity (< 1 µm) (Figure 1D, median ribbon-to-mitochondria distance = 174 nm, n = 17 out of 21 ribbons).”

Materials and methods – “A subset of measurements were taken from more than one ribbon within a hair cell. Near 81% of ribbons, a mitochondrion could be clearly identified within 1 µm of a ribbon (17 out of 21 ribbons).”

Next they claim that in lateral-line hair cells, mitochondria are present near ribbons. However, the images shown clearly reveal that mitochondria are everywhere in hair cells, which unsurprisingly will include the areas near ribbons. Therefore, the claim that mitochondria are poised to impact ribbon synapses is overreaching.

We have toned down our statement.

Results – “Overall our TEM and Airyscan imaging suggests that in lateral-line hair cells, mitochondria are present near ribbons.”

The data shown cannot be used to claim that "MitoGCaMP3 signals propagated apically within the mitochondria". Instead, MitoGCaMP3 signals propagated apically. The use of Ru360 does not directly support the claim either, because the authors would be blocking Ca^2+^ entry into mitochondria generally from any source including cytoplasm or the ER, and not a spread of within the mitochondria.

We agree that the data on mitochondrial calcium uptake and pharmacological block of calcium channels cannot distinguish between propagation of calcium within mitochondria versus a gradient increase in calcium level at the cytosol that drives the correspondingly synapse-to-apex direction of mitochondrial calcium uptake. Results, "MitoGCaMP3 signals propagated apically within the mitochondria" is corrected to "MitoGCaMP3 signals propagated apically”.

I am surprised with what appears to be zero MitoGCaMP3 signal outside the active-zone area at the start of the 2-s mechanical stimulation of neuromasts, given the long perdurance of the signal after stimulation (5 min to return to baseline). Hair cells must have been stimulated constantly during the experiment. Otherwise, the authors may want to explain this observation and how the prevented hair-cell activity before the 2-s evoked stimulation.

It is true that after stimulation the MitoGCaMP3 signal took nearly 5 min to return to baseline. Because of this slow return, we did wait 5 min between each evoked trial for consistency. This is now reflected in the Materials and methods – “Due to the slow mito-Ca^2+^ return to baseline after stimulation (~5 min), we waited a minimum of 5 min before initiating a new evoked GCaMP6sCAAX or MitoGCaMP3 acquisition.”

In addition, we added a piece of data to indicate that the mitochondria can still take up Ca^2+^ even if the MitoGCaMP3 has not returned to “pre-stimulus baseline”. This data is a 2-s stimulus repeated every 10s (Figure 1—figure supplement 1D). Although smaller, additional mito-Ca^2+^ uptake is possible with subsequent stimuli. Results – “Despite this long time-course of recovery to baseline, we were still able to evoke additional rises in MitoGCaMP3 signal 10 s after stimulation (Figure 1—figure supplement 1D, ∆F/F_0_).”

With regards to the MitoGCaMP3 baseline at zero prior to stimulation – there is definitely some amount of Ca^2+^ in mitochondria at rest. Otherwise it would be not be possible to see the structures prior to stimulation. The heat maps in Figure 1E are just helpful way visualize the relative change (∆F) in MitoGCaMP3 from ‘rest’ in on a pixel-by-pixel basis. The graphs in E’-E’’ are normalized to start at MitoGCaMP3 measurements at zero in all ROIs (∆F/F_0_). We have added ∆F and ∆F/F_0_ to the text to help clarity what changes in MitoGCaMP3 we are referring to. We have also added additional information regarding what our plots and heatmaps represent in the Materials and methods.

Figure 3F: Are Ru360 2 μM for 1hr and DMSO for Ru360 10 μM for 1hr significantly different?

Yes, these measurements are significantly different. Although our overall results are consistent over time, there are slight differences between immunostains, clutches, as well as modifications to our imaging system over years that are challenging to control. Because of this we only compare our experimental data to control samples from the same clutch and immunostain. We have added a statement to our Materials and methods emphasizing this – “Due to slight variability between clutches and immunostains we only compared experimental data taken from the same clutch, immunostain and imaging session.”

The authors measure and compare Ribeye b area assuming that the 3D shape of the ribbons is identical under every treatment. Have they confirmed the case?

This is a great point. We have done circularity measurements on the areas we measure for each treatment. The lower concentration of Ru360 (2 µM) does not impact ribbon circularity, but longer, 1 hr application of 10 µM Ru360 does change ribbon circularity. We have considered making 3D-volume measurements, but our microscope does not have sufficient Z-axis resolution to reliably predict these relatively small volumes. Even the X-Y dimensions we are measuring near the limit of our imaging system. The 3-D volumes would be relative estimations skewed in Z.

Within our ribbon dataset we do reliably capture ribbons from both the side and the top. Because we are viewing ribbons from multiple angles, we feel that our max-projected area measurements are the best approximation of ribbon size.

I am not sure what the authors refer to with "spontaneous Ca_V_1.3 and MCU Ca^2+^ activities".

We have edited the text to show that we are referring to Ca^2+^ uptake in developing hair cells. Results – “Our Rex-YFP measurements suggest that in developing hair cells, Ca_V_1.3 and MCU Ca^2+^ activities normally function to decrease the NAD^+^/NADH ratio; […]”

I am surprised that the authors do not discuss their findings in relation to the findings of Zenisek and Matthews, 2000.

We have now included Zenisek and Matthews, 2000, in our Discussion – “Study of synaptic mitochondria at ribbon synapses in retinal-bipolar cells found that mito-Ca^2+^ uptake was sporadic and did not significantly contribute to the time course of evoked presynaptic-Ca^2+^ responses or Ca^2+^ clearance (Zenisek and Matthews, 2000). This work concluded that mitochondria may contribute indirectly to Ca^2+^ clearance from the synaptic terminal by providing ATP to fuel the PMCA.”

In subsection “Functional significance of ribbon size”, the authors state: "Functionally, compared to smaller ribbons, larger ribbons are associated with afferent fibers with less spontaneous activity and higher thresholds of activation". What fibers are the authors referring to? Afferent fibers of the lateral line do not show intrinsic spontaneous activity. Their spontaneous activity is given by spontaneous release of glutamate from the hair cells.

We have modified the text to be clear that we are referring to auditory inner hair cells for this comparison. Discussion – “Similarly, in the mammalian auditory system, ribbon size is correlated with differences in afferent activity. […] Compared to larger ribbons, smaller ribbons within inner hair cells are innervated by afferent fibers with higher thresholds of activation and lower rates of spontaneous activity.”

Is there any indication that altered Ca^2+^ entry into mitochondria alters the localization of Ca_V_1.3 channels?

We have now added data to quantify the localization of Ca_V_1.3 channels in mature hair cells after blocking mito-Ca^2+^ uptake. We find that while blocking calcium entry into mitochondria with 2 μm Ru360 depresses presynaptic Ca^2+^ responses, Ca_v_1.3 channels are still present at synapse (Figure 2E-G).We also see a significant increase in Ca_V_1.3-channel density at synapses after MCU block (Figure 2H). Results – “After the 1-hr 2 µM Ru360 treatment, Ca_V_1.3 clusters were still present at synapses, but the channels were at a significantly higher density compared to controls (Figure 2E-H). These findings indicate that in mature hair cells, partial MCU block may impair presynaptic function by altering Ca_V_1.3 channel density.”

This result is also included in our Discussion – subsection “Role of evoked mito-Ca^2+^ uptake in mature hair cells”.

The authors must discuss their findings in relation to Lv et al., 2016.

Good point, we have now added the findings of Lv et al., 2016 to our Discussion – “Work in zebrafish has characterized lateral-line hair cells largely depleted of full-length Ribeye (Lv et al., 2016). […] If full-length Ribeye and its NAD(H) binding domain is the site of regulation, we predict that Ribeye-depleted hair cells will be unaffected by perturbations in mito-Ca^2+^ and NAD(H).”

Reviewer #2:Beurg et al., 2010, provide evidence for mitochondrial responses to apical calcium entry in cultured mammalian hair cells. These results contrast to the findings of the current work, where calcium spreads from basal mitochondria and are blocked by the calcium channel blocker isradipine. Is this due to differences in the kinetics of stimulation or differences in the type of hair cells?

This is an important comparison to make. We do occasionally observe apical mito-Ca^2+^ uptake that is distinct from the mito-Ca^2+^ uptake at the base of the hair cell. We have not fully characterized these responses for reproducibility or their pharmacological properties. In our revision we have found that blocking mito-Ca^2+^ uptake (via MCU or VDAC block) actually increases mechanotransduction (see below and Figure 2—figure supplement 1A-B’). This supports the work of Beurg et al., 2010, that suggest apical mitochondria may be acting as a buffer to reduce responses in the apical part of the hair cell. We feel this result definitely warrants future exploration, but is outside the current scope and focus of this manuscript. But, in the Discussion we now state that blocking mito-Ca^2+^ uptake increases mechanotransduction and compare this result to Beurg et al., 2010.

Discussion – “In mammalian auditory hair cells, mito-Ca^2+^ uptake has been observed to buffer Ca^2+^ beneath mechanosensory hair bundles (Beurg et al., 2010). […] In the future it will be extremely interesting to explore the role apical mitochondria play in mechanotransduction.”

Results displayed in Figure 2 are described as in mature hair cells but it appears some of the data were from 3 dpf. It is not clear why immature cells are included given the later results showing age-dependent differences.

We agree that including an overall immature cell population in Figure 2 when that focus is on mature hair cells is confusing. Although there are a subset of synaptically active mature hair cells at 3 dpf, for clarity, we now only include mature hair cells at 5 dpf in this figure. Figure 2 has been modified to show MitoRGECO1 and GCaMP6sCAAX in mature hair cells at 5 dpf.

It is somewhat surprising that blocking calcium uptake into mitochondria reduces presynaptic calcium signals. It is potentially possible that Ru360 may have non-specific effects such as blocking mechanotransduction. This could be tested by examining apical calcium signals, which the Kindt lab has found occur in all hair cells in response to mechanotransduction. An alternative inhibitor of mitochondrial calcium uptake such as VDAC inhibitors might also strengthen this result.

We have used an alternative inhibitor of mitochondrial calcium uptake. In addition to using the MCU blocker Ru360, we have added data using the VDAC inhibitor TRO 19622. Similar to Ru360, TRO 19622 blocks mito-Ca^2+^ uptake. Results – “We confirmed these results by applying TRO 19622, an antagonist of the voltage-dependent anion channel (VDAC). VDAC enables transport of ions including Ca^2+^ across the outer mitochondrial membrane (Schein et al., 1976; Shoshan-Barmatz and Gincel, 2003). We observed that similar to the MCU antagonist Ru360, a 20-min treatment with the VDAC antagonist TRO 19622 also impaired stimulus-evoked MitoGCaMP3 signals (10 µM TRO 19622, Figure 1—figure supplement 1E).”

Similar to Ru360,TRO 19622 also blocks presynaptic Ca^2+^ responses in mature hair cells. Results – “A similar reduction in GCaMP6sCAAX signals were observed after a 20-min application of the VDAC inhibitor TRO 19622 (Figure 2—figure supplement 1D-E’, 10 µM TRO 19622).”

Ru360 is an analog of Ruthenium Red, a known mechanotransduction channel blocker. Therefore, as suggested, we have tested the apical calcium signals after MCU block (Ru360) and VDAC block (TRO 19622). We found that neither drug blocks mechanotransduction.Results – “Previous work in zebrafish-hair cells demonstrated that isradipine specifically blocks Ca_V_1.3 channels without impairing mechanotransduction (Zhang et al., 2018). […] Neither compound blocked mechanotransduction (Figure 2—figure supplement 1A-B’).”

Throughout the authors use the term "formation" to describe effects on synapses. It might be slightly pedantic, but it appears that the effects they observe do not affect formation (no changes in the number of synapses; Figure 5D, 7D) but rather size and shape (Figure 5E, 7E). Formation implies the localization of new synapses rather than alteration of synapses already formed by other mechanisms.

We agree that it is a bit misleading to use the term formation, because none of our treatments prevent synapses from forming. Rather we are studying an aspect of formation. Where we were able, we have changed “formation” to “size” in the Abstract and text body.

It is puzzling that blocking mitochondrial calcium accumulation results in an increase in NAD^+^. Mitochondrial oxidation of NADH to promote NAD^+^ production is usually thought to be stimulated by calcium, which is the opposite effect of the one observed here. Given the puzzling nature of the link between calcium and NAD^+^, could additional experiments be performed to strengthen the results? Could uncoupling of mitochondria have an effect on ribbon size? Could pyruvate accumulation be the result? Could similar effects be seen with the MPC inhibitor UK5099?

Unfortunately, the relationship between NAD^+^/NADH and mitochondria calcium in neurons or hair cells is not straightforward. In our Discussion we have tried to draw comparisons between our observed results and studies in cardiac myocytes where this relationship has been most widely studied. Discussion – “Some insight into these differences can be inferred from cardiac myocytes where the relationship between mito-Ca^2+^ and NAD(H) redox has been extensively studied. […]These results are consistent with the changes in NAD(H) redox we observe in developing, but not mature hair cells.”

We also agree that additional pharmacology could be used to understand more precisely how cellular metabolism and mitochondria calcium impact ribbon size. Unfortunately, many of the mitochondrial uncouplers and metabolic inhibitors are likely to be quite toxic to larval zebrafish. And careful pharmacology is considerably time consuming. In the past 2 months we have used much of this time to confirm many of our Ru360 results with the VDAC inhibitor TRO 19622. Additional pharmacology will be extremely interesting, but may be beyond the scope of this present study.

Reviewer #3:[…]Subsection “Mito-Ca^2+^ uptake at ribbons is MCU and Ca_V_1.3 dependent”, last paragraph: The authors claim that the effect of Ru360 is dose-dependent. Were other doses used? If so, what was the cut-off dose for the effect, i.e., what was the IC_50_ or EC_50_?

In our original submission, only 2 and 10 µM Ru360 were presented. We have added other doses to more accurately estimate the IC_50_. The IC_50_ is included in the Results – subsection “Mito-Ca^2+^ uptake at ribbons is MCU and Ca_V_1.3 dependent”, third paragraph, and the dose response curve is plotted in Figure 1—figure supplement 1F.

Subsection “Mito-Ca^2+^ uptake occurs in cells with presynaptic Ca^2+^ influx”, first paragraph: It has been described that approximately 10-40% of PSDs in mammalian hair cells do not have ribbons and about a third of boutons do not have ribbons (Sadeghi et al., 2014; Lysakowski and Goldberg, 1997). This is roughly equal to the 40% of cells in the present paper in which mito-Ca^2+^ uptake was present and the 30% of cells in the Zhang et al., 2018, paper that are synaptically active. I wonder if this is a "snapshot in time" of ribbon turnover, which has been documented in pinealocytes, but has not so far been described in hair cells. There were, in fact, a couple of papers on circadian rhythms affecting ribbon numbers in cochlear hair cells (Swetlitschkin and Vollrath, 1988; Molina Mira and Martinez-Soriano, 1991) cited in this review (Wagner, Cell Tissue Res., 1997, PMID: 9027298), which will likely be interesting to the authors, as the review also discusses developmental changes in ribbon synapse production due to Ca^2+^ effects.

It is true that pre- and post-synapses were present in active and inactive cells in Zhang et al., 2018. In this study we also added quantifications showing Ca_V_1.3 clusters at all synapses after MCU block in mature hair cells (Figure 2E-H). And yes – these images are merely a snapshot in time. Synaptic structures are likely dynamic, although currently how dynamic is unclear. We are presently trying to study synaptic structures, in hair cells, in vivo! Therefore, we now make reference to this in our Discussion – “How could synapses be changing in mature hair cells during this pathology? […] In the future, live imaging studies will help resolve whether there are different mechanisms underlying ribbon enlargement in mature and developing hair cells.”

Subsection “Evoked mito-Ca^2+^ uptake is important for mature synapse integrity and cell health”, last paragraph: This study was done in posterior neuromasts. I wonder if it is known whether there are differences between anterior and posterior neuromasts in terms of sensitivity or ribbon size?

Unfortunately, there are not many studies that make direct comparisons between anterior and posterior neuromasts. To be more comprehensive, we have reexamined our slides and quantified synapses in anterior lateral-line neuromasts. We find that the ribbons and PSDs are similar in size between anterior and posterior lateral-line neuromasts (Figure 3—figure supplement 1A-B, Figure 5—figure supplement 1A-B). In addition, Ru360 enlarges ribbons in developing hair cells within the anterior-lateral line (Figure 5—figure supplement 1C). We also found that anterior lateral-line neuromasts have robust evoked mito-Ca^2+^ uptake (Figure 1—figure supplement 2). Overall these results suggest that anterior and posterior neuromasts have synapses of similar sizes, evoked mito-Ca^2+^ uptake and that mito-Ca^2+^ plays a role in regulating ribbon size in both populations of developing hair cells.

I also wonder if the ribbons are becoming bigger to compensate for the loss of synapses. In the microgravity of spaceflight, ribbons can become more numerous, perhaps compensating for the loss of gravity (Ross, Acta Otolaryngol., 2000) or less numerous (Sultemeier et al., 2017).

We have added the above alterative to our discussion of how ribbons in mature hair cells can get larger. See above.

Subsections “Spontaneous mito-Ca^2+^ uptake regulates ribbon formation”, second paragraph and “Role of spontaneous mito-Ca^2+^ uptake in developing hair cells”: The authors state that in developing hair cells, after application of Ru360, there is an increase in ribbon size. There is also, apparently, an increase in the number of cytoplasmic Ribeye-positive inclusions. How do we know that the size increase is not just that the ribbon clusters are not separating? Normally, ribbons are produced and "float" within the cytoplasm in clusters, presumably being transported through the cytoplasm by microtubules (this is supported by the observable Ribeye-positive cytoplasmic inclusions), then they separate into individual ribbons near and at the plasma membrane. Again, this may only be discriminable by TEM.

It is true that we cannot tell why ribbons or Ribeye aggregates are becoming larger in developing hair cells. But we would like to know! We have added a statement about these aggregates to our Discussion – “We also observed an increase in cytoplasmic Ribeye aggregates after MCU block (Figure 5F-G). […] NADH(H) redox could alter Ribeye interactions and alter the overall accumulation or separation of Ribeye within aggregates or at the presynapse.”

Subsection “MCU and Ca_V_1.3 channel activities regulate subcellular Ca^2+^ homeostasis”, last paragraph: Another possibility that doesn't seem to be considered in the paper or in the Discussion is that there are cytoplasmic calcium binding proteins that also buffer the Ca^2+^.

We now mention calcium buffering proteins in our discussion of buffering the hair-cell cytosol – Discussion – “But similar to work on retinal-bipolar cell ribbons, blocking mito-Ca^2+^ uptake did not raise cyto-Ca^2+^ levels, indicating it may not be critical for Ca^2+^ clearance (Figure 6A-C). Instead, cyto-Ca^2+^ levels may be maintained by the PMCA, along with the numerous Ca^2+^ buffering proteins that have been identified in hair cells (Hackney et al., 2005; Haeseleer et al., 2000; Steyger et al., 1997).”

Subsection “Functional significance of ribbon size”, second paragraph: The synaptic ribbons in central/striolar type II hair cells feeding into irregular, dimorphic vestibular afferents are larger than those feeding into peripheral regular, dimorphic afferents (Lysakowski and Goldberg, 1997). Similarly, irregular afferents have lower spontaneous background activity than regular afferents (Table 2, Goldberg, Smith and Fernández, 1984). So, there is some similarity to cochlear afferents.

Great suggestion! The vestibular hair-cell synapse and afferent have been added to our Discussion – “In many instances ribbon size is correlated with functional properties of the synapse. […] Functionally, afferents that innervate hair cells with larger ribbons in the extrastriolar region have higher activation thresholds and lower rates of spontaneous activity compared to afferents that innervate the striola region (Eatock et al., 2008; Goldberg et al., 1984; Risner and Holt, 2006).”